# The Sharklogger Network—monitoring Cayman Islands shark populations through an innovative citizen science program

Johanna Kohler [1,2,3]*, Mauvis Gore [2,3], Rupert Ormond[2,3], Timothy Austin[1], Jeremy Olynik[1]

1 Department of Environment, Cayman Islands Government, George Town, Grand Cayman, Cayman Islands, 2 Centre for Marine Biodiversity and Biotechnology, Heriot-Watt University, Edinburgh, Scotland, United Kingdom, 3 Marine Conservation International, Edinburgh, Scotland, United Kingdom

* johanna.k.kohler@gmail.com

## Abstract

The use of citizen science can be a cost-effective tool in conservation science but mostly relies on the collation of incidental sighting reports. This study describes the design, operation, and findings of a novel, closely-guided citizen science program (the 'Sharklogger Network') in the Cayman Islands. Participants from the recreational SCUBA diving community used a standardized, effort-based protocol to monitor local coastal shark populations. Over two years (2017–2018) a total of 69 participants conducted 24,442 dives across 472 dives sites and recorded 4,666 shark sightings from eight shark species, of which Caribbean reef shark, nurse shark, and hammerhead spp. were the most frequently observed and encountered throughout the year. The data from dive logs provided evidence for species-specific distribution and abundance patterns across and within islands, indicating a greater abundance of sharks in areas with less anthropogenic activity and with a greater exposure to strong currents, regardless of whether the area was an MPA or not. While both Caribbean reef sharks and nurse sharks showed species-specific depth and habitat preferences, the recording of recognizable individuals showed that some individuals of both species have relatively small home ranges and high site-fidelity to specific areas. The study also provided the first confirmation of reproductive behaviour in both Caribbean reef and nurse sharks taking place in summer (May–August). Experience showed that along with generating valuable data the program, by engaging local stakeholders, also enhanced public awareness of shark conservation issues. This study demonstrates that this citizen science methodology can be an affordable and non-invasive tool for the reliable long-term monitoring of shark populations.

## Introduction

Recent studies have highlighted the global decline of reef sharks [1–5]. In Cayman, the shark populations are believed to have decreased in the 1960's because of its historic shark fishery [6,7]. However, due to the extensive evidence of the beneficial role that sharks play in marine ecosystems [8–11], the conservation of Cayman's sharks has become a priority [12,13]. To

**Data availability statement:** All relevant data are within the paper and its Supporting Information files.

**Funding:** This work was supported through funding generated by the sales of the Whitetip Lager from the Cayman Islands Brewery Ltd. (2017-2018) (DoE) www.cib.ky/caybrew. A small proportion of the income from the sale of the local brewery's 'Whitetip Lager' is donated to the Cayman Islands Department of Environment's shark research and provided additional funding for this study. The funders had no role in study design, data collection and analysis, decision to publish, or preparation of the manuscript.

**Competing interests:** The authors have declared that no competing interests exist.

promote the sustainability of marine resources Cayman has a long-established network of Marine Protected Areas (MPAs; est. 1986, totaling 55% of the coastal shelf) and added the nationwide protection of all shark species in 2015 [13]. Nevertheless, the human population of the small and relatively remote island-nation continued to increase in the last decade [14] and has resulted in major development of the coastline, particularly on Grand Cayman [15]. This development led, during the 1990s and early 2000s, to a noticeably degraded coastal environment and increased pollution around Grand Cayman, including some loss of shallow-water habitats, such as mangrove forest [16–18]. The three islands are a popular tourist destination because of their clear, warm water, and relatively healthy reefs [19,20]. Since the first cruise ship arrived in Cayman in 1937, the number of cruise ships arriving in Grand Cayman has increased to an average of 615 per year, bringing about 1.5 million passengers per year (1996–2018 [21]). In-water activities such as SCUBA diving, sport fishing, and visits to "Stingray City" (a sandbar in the middle of North Sound, which is the number one tourist attraction in Cayman [12,22] are popular among tourists and residents. In the face of these activities knowledge of how sharks use coastal waters in the Cayman Islands has become critical for their effective conservation and management, since such anthropogenic disturbances have been observed to impact sharks in other regions [16,20,23–27].

In the central Caribbean, historic catch records and single sightings of species provide some indication of the diversity and abundance of shark species [6,28,29]. To provide scientific-based information to local policy makers, organized shark research in Cayman commenced in 2009 [12]. Taken together, preliminary results provided insight into the diversity, relative abundance, and specific habitat use of species in Cayman [12,30] which seem similar to that of reef sharks in the wider Caribbean [31–37]. However, due to challenging logistics and limited financial resources available for scientific research efforts, distinct knowledge gaps in the ecology of sharks in Cayman persist. Since the MPAs in Cayman were originally designed as much to protect areas with high diving pressure as to strategically protect biodiversity (TA and JO per. obs.), it has also become important to assess whether existing MPAs are benefiting local shark populations, an issue that has also arisen elsewhere [23,38–40].

As most high-resolution and long-term marine research is resource intensive, it is impracticable for most conservation agencies to collect the amounts of data ideally required. Citizen science, the use of volunteers recruited from the general public to collect data, can be a cost-effective research tool and is now widely used in marine conservation studies [41–44], including those on sharks [45–48]. Such data can however have various sources of bias resulting from such factors as inadequate training of participants, use of non-standardized protocols, insufficient volunteer participation, and detection bias [49–52]. Furthermore, as frequently emphasized in monitoring recommendations, the count of zeros (i.e., zero-data) is necessary to measure survey effort and permit effective data analyses [48,53–56]. In contrast most previous citizen science programs using recreational SCUBA divers to monitor sharks recorded only positive sightings of sharks [45,46,57,58]. Due to the lack of zero-data, most of such programs have value only as pilot studies [45,49,59] or to supplement more detailed scientific research [60–65]. As citizen science becomes more widely used, it is important to design programs that gather high quality data [50,51,66].

The Cayman Islands offer various features which make it a suitable location to test the monitoring of sharks by citizen science. Firstly, with about 25 dive operators catering for visitors, and with hundreds of residents SCUBA diving regularly, there is potential for sufficient participation to ensure adequate data [52,59,67]. Secondly, the clear water and relative constant diving activity throughout the year make sustained monitoring feasible and limit the issue of detection bias encountered by others [68–72]. Thirdly, the nation-wide ban on feeding wildlife (with the exception of the Wildlife Interaction Zone (WIZ), where Southern Stingrays

(*Hypanus americanus*) may be fed) limits potential bias due to artificial food provisioning of sharks, such has been employed in some other studies [73–76].

In 2016, the launch of the 'Sharklogger Network' aimed to involve the majority of the local recreational SCUBA diving community in a closely-guided and standardized citizen science program to assist with the monitoring of local shark populations. While the benefits of a standardized sample protocol, the recording of zero data and guidance of participants have previously been demonstrated in relation to shark citizen science projects [48,54,77,78], none of these studies monitored shark populations continuously for more than four consecutive months or over multiple years. Whilst White et al. [54] and Sattar et al. [78] systematically collected long-term data (two decades: 1993–2013, 4 years: 2009–2013) via SCUBA divers, generating relatively large databases (27,527 dives, 11,704 dives respectively), both programs collaborated with only one or a few dive operators. Seguigne et al. [48] recruited multiple divers and dive centers across French Polynesia to record long-term data (6.5 years: 2011–2018: 13,916 dives and 114 volunteers) but these dive centers were actively feeding sharks on some dives. The program in this study is novel in collecting data with high spatiotemporal resolution over multiple years, in using mass participation of volunteer divers, and in being relatively non-invasive by recording the natural presence of sharks without the use of bait.

The aims of the present study were to: (1) assess whether the local recreational dive community could be used to monitor coastal shark populations, (2) assess the species composition, relative abundance, spatial distribution, and behaviour of sharks in the Cayman Islands, (3) identify environmental drivers of local shark abundances, and (4) enhance public awareness by engaging local stakeholders in practical shark research. The following hypotheses were considered: (1) that among a number of coastal shark species, highly reef-associated species such as the Caribbean reef (*Carcharhinus perezi*) and nurse shark (*Ginglymostoma cirratum*) are most abundant, (2) that the relative abundances of reported species were different between islands, areas on each island, and different months but not between years, (3) that environmental drivers such as temperature, depth and current strength influence local sharks, (4) that the behaviour of sharks is species-specific, (5) and that individual home ranges cover multiple dive sites (approximately hundreds of meters) in the Cayman Islands.

## Materials and methods

### Study area

Located in the western Caribbean Sea, the Cayman Islands, Grand Cayman (19.3222° N, 81.2409° W), Little Cayman (19.6897° N, 80.0367° W), and Cayman Brac (19.7235° N, 79.8017° W) (Fig 1), have a total land area of 264 km² [79]. Little Cayman and Cayman Brac, are smaller than Grand Cayman and known as the "Sister Islands", are approximately 6.5 km apart from each other and lie approximately 110 km to the northeast of Grand Cayman. The narrow coastal shelf (maximum width: 1.5 km) around each island consists of shallow near shore lagoons, two reef terraces (shallow: 5–15 m, deep: 16–25 m) [79,80], and almost vertical slopes extending to more than 2,000 m [79]. The benthic marine environment comprises of seagrass, sandy bottom, coral rock, and coral reef, providing habitat for numerous reef fish [29,79,80]. For each island, due to the prevailing easterly approach of storms and waves, the exposure of the coast ranges from low energy on the leeward margin (i.e., west coast), through medium energy (i.e., north and south coast) to high energy on the exposed, windward margin (i.e., east coast).

The underwater conditions are characterized by occasional, strong oceanic currents [79], relatively warm (mean 28.2 ± 1.1 ºC) sea surface temperatures (SST) with only small changes through the year [range = 26–30.5 ºC; [81]], and good visibilities (up to > 40 m). During

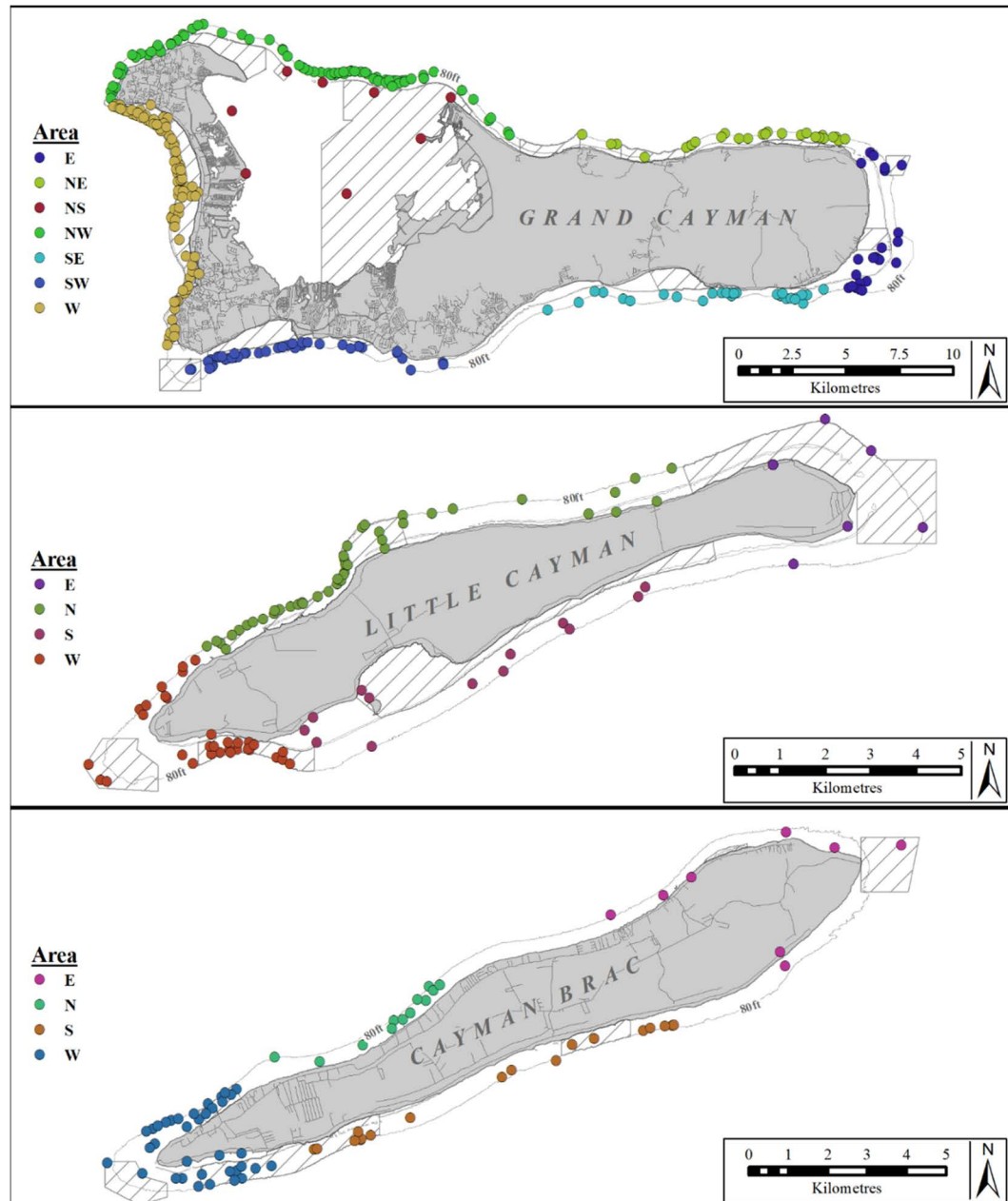

**Fig 1. Map showing the location of dive sites accessible by boat or from the shore in the Cayman Islands.** The color of circles indicates dive sites allocated to the different sectors into which the study area was divided for statistical purposes. The line around each island indicates the 25 m (80 ft) depth contour and the shaded areas indicate the extend of MPAs. Created by the Department of Environment, Cayman Islands Government. Insert layer's geography was developed by Esri and sourced from Garmin International, Inc., the U.S. Central Intelligence Agency (The World Factbook), and the National Geographic Society for use as a world basemap [82].

the period of this study, Marine Protected Areas (MPAs) covered 45% of the coastal shelf of Grand Cayman, 48% of that of Little Cayman, and 25% of that of Cayman Brac ([13] Fig 1). More recently, MPAs have been extended to cover 55% of the total coastal shelf and restrictions on fishing increased [13]. However, the data in this study were collected before this enhancement, so that only the earlier regulations are taken into consideration here. Across the

three islands, boat diving is accommodated through 375 public moorings while shore diving has numerous public shore access points (Fig 1). The regulations allowed diving inside most MPAs and fishing both from shore and beyond the 25 m depth contour.

## Data collection

All data on sharks and environmental variables reported here were recorded by citizen science volunteers, who include both professional dive center staff and local recreational divers, as well as a small number of local snorkelers, recruited to the Sharklogger Network. Records of shark abundance (number of sharks observed during a dive, including zero sharks), notable shark behaviour, and environmental variables were collected by participants on all three islands, more or less daily, throughout the year, and reported on a standardized data sheet ('Shark Log', Fig 2), developed by one of the authors (JK). Participants were unpaid and conducted dives at their own responsibility. With the submission of a Shark Log, participants gave their consent for the use of the data by the program. Regardless of dive equipment (SCUBA or snorkel) used, participants in the program are hereafter referred to as 'divers' and sampling events as 'dives'.

## Recruitment of divers

Divers were recruited to the program by one of the authors (JK) who acted as the island-based coordinator. At the launch in 2016, and thereafter, the initiative was publicized through local media including TV, radio, newspapers, and social media, to encourage volunteer recruitment. The public was asked to contact the coordinator via email or social media. Additionally, the coordinator contacted specific dive centers and individual divers that were known to have a positive attitude towards shark conservation and to dive frequently. No selection was made with regards to occupation, gender, age, or education level of divers and volunteers were considered to have been recruited to the program once they had submitted their first Shark Log.

| DATE (DD/MM/YYYY) | DISTRICT or PART OF ISLAND (Name/N/NE/E etc.) | DIVE SITE (if it has established name) | TIME OF DAY (approx.) | DIVE TIME (approx.) | MAX. DEPTH (approx.) | TEMPERATURE (Deg C) | while Lionfishing? | VISIBILITY (to nearest 5m/15ft) | CURRENT (weak 0 - strong 3) | NO. of SHARKS SEEN (state if ZERO!) | SHARK SPECIES (if known) e.g. Caribbean reef, blacktip, nurse shark | SIZE (approx. to nearest 10cm/4inch) | Sex (female / male / unknown) | TAGGED / NAME / MARKINGS? (Tag no. / colour / scars / hooks) | No. of Divers / Guests (if applicable) | Comments |
|---|---|---|---|---|---|---|---|---|---|---|---|---|---|---|---|---|
| 01-Apr-16 | West Side | Darvin's Theory | 10:00 | 55 | 90 | 80 | no | 80 | 0 | 0 | | | | | 3 | |
| 01-Apr-16 | West Side | Round Rock | 11:30 | 45 | 80 | 80 | no | 100 | 0 | 1 | Nurse shark | 4ft | female | no tag, hook in left corner of mouth | 15 | followed us the entire dive |
| 01-Apr-16 | West Side | Devil's Grotto | 13:30 | 50 | 40 | 79 | yes | 80 | 2 | 0 | | | | | 15 | |
| 02-Apr-16 | West Side | Meridian Drop Off | 09:30 | 50 | 90 | 79 | no | 90 | 0 | 0 | | | | | 8 | |
| 02-Apr-16 | West Side | Lone Star Ledges | 11:30 | 50 | 50 | 80 | no | 90 | 0 | 0 | | | | | 8 | |
| 02-Apr-16 | North Wall | Lemon wall | 02:30 | 45 | 50 | 80 | no | 100 | 0 | 0 | | | | | 3 | |
| 02-Apr-16 | North Wall | Eagleray Pass | 10:00 | 50 | 80 | 80 | no | 100 | 1 | 0 | | | | | 3 | |
| 02-Apr-16 | North Wall | Durgon's Domain | 10:00 | 45 | 80 | 80 | yes | 80 | 0 | 0 | | | | | 12 | |
| 02-Apr-16 | North Wall | Lemon reef | 12:30 | 45 | 40 | 80 | no | 80 | 0 | 2 | Nurse shark | 4ft + 7ft | unknown + female | no tags | 12 | small shark was sleeping under reef ledge |
| 03-Apr-16 | West Side | Round Rock | 10:00 | 45 | 80 | 80 | no | 80 | 0 | 0 | | | | | 14 | |
| 03-Apr-16 | West Side | Eagle's Nest | 10:30 | 45 | 80 | 60 | no | 80 | 0 | 0 | | | | | 15 | |
| 04-Apr-16 | South | Black Forest | 09:30 | 45 | 80 | 81 | no | 90 | 0 | 0 | | | | | 20 | |
| 04-Apr-16 | South | Laura's Reef | 11:30 | 50 | 50 | 80 | no | 80 | 2 | 0 | | | | | 20 | |
| 04-Apr-16 | West Side | Disneyworld | 14:30 | 50 | 100 | 81 | no | 90 | 0 | 1 | Caribbean reef shark | 6ft | male | yes, didn't see tag number | 3 | shy and cautious of divers |
| 04-Apr-16 | West Side | Kittiwake | 10:00 | 45 | 80 | 80 | no | 90 | 0 | 0 | | | | | 14 | |
| 04-Apr-16 | West Side | O.V. Wreck | 11:30 | 50 | 50 | 81 | no | 90 | 0 | 1 | Nurse shark | 7ft | male | Finn (hook scar on mouth) | 14 | very friendly |
| 04-Apr-16 | West Side | Darvin's Theory | 12:30 | 45 | 80 | 80 | no | 90 | 0 | 0 | | | | | 17 | |
| 04-Apr-16 | West Side | O.V. Wreck | 14:00 | 50 | 50 | 80 | no | 90 | 0 | 0 | | | | | 17 | |
| 05-Apr-16 | West Side | O.V. Wall | 09:45 | 45 | 80 | 80 | no | 80 | 0 | 0 | | | | | 9 | |
| 05-Apr-16 | North Wall | Tarpon Alley | 11:00 | 50 | 50 | 80 | no | 80 | 0 | 0 | Caribbean reef shark | 7ft | female | scar on dorsal from tag | 9 | Name "Spot" |
| 05-Apr-16 | North Wall | Creole Cliff | 14:30 | 60 | 50 | 80 | no | 70 | 1 | 0 | | | | | 2 | |

**Fig 2. Part of the example of a completed Shark Log provided by the coordinator to participating divers.** A full copy can be viewed on the last page of S1 Fig.

## Training of divers

Divers were trained remotely through written material including relevant information and the sample protocol (detailed below) provided as part of the 'introduction package' (S1 Fig). Training especially emphasized the necessity of recording 'zero data' (data on dives without any shark encounter), shark species identification (highlighting common local misidentifications), shark sex determination and size estimation, the recording of dorsal fin tags, and potential behaviour of interest. Additionally, to ensure correct data entry, an example of a completed Shark Log (S1 Fig) was provided. It was assumed that the training of a diver was successful if no further questions were asked when the volunteer was next contacted. If feedback was given, the coordinator ensured that questions were clarified, and that the sample protocol was understood before data collection by that individual commenced.

## Programme management

Management of the Sharklogger Network included continued engagement with and guiding of participants. Two weeks after the delivery of the introduction package, relevant divers were contacted by the coordinator to (1) confirm that the training was successful, (2) confirm that the diver had started data collection, and (3) assess whether additional support (for example: dives with the coordinator, in person visits, presentations, explanations, or other guidance) was required.

To ensure volunteers submitted their monthly data, at the end of each month, data (i.e., Shark Log) submission was prompted through a 'Call Shark Log' email. Divers emailed copies of their Shark Log as either an Excel file, typed PDF, or a picture (jpeg, png) of handwritten original records (S2 Fig). Typed PDF files were converted into Excel format, and pictures of handwritten Shark Logs manually entered into an Excel file by the coordinator and volunteers from the public. Data transferred by volunteers were confirmed by the coordinator.

Divers that failed to submit a Shark Log for six months received a 'Bounce Sharklogger' email, asking for confirmation of continued participation in the program.

Communication between the coordinator and divers was promoted through monthly newsletters, distributed as part of the Call Shark Log-email; these included relevant local shark news, such as interesting observations from submitted Shark Logs (e.g., rare sightings or injuries of individual sharks), exclusive fieldwork insights and preliminary results of ongoing shark research [12,83–86].

## Sample protocol for dive and shark data

Divers were asked to conduct their dives normally, within their specific sport diver limits and using a suitable dive profile (depending on their personal experience, air consumption, ease and comfort in water, and fitness). The analysis reported here includes data from all reported dives undertaken between 1st January 2017 to 31st December 2018, including dives when divers did not encounter any sharks. For each dive, divers recorded: the date (dd/mm/yy), the time of day, the dive location, the dive duration (min), the maximum depth (m), the water temperature (°C), the visibility (m), dive group size (number of divers). Dive duration, water temperature and maximum depth were recorded from dive computers. In case a diver did not dive with a personal dive computer, or the relevant information was not stored on their device, data was taken from the dive buddy. Lionfish culling was defined as dives with at least one lionfish kill. Dives on which divers intending to cull (i.e., carrying spears) but no kill occurred (due to either a lack of lionfish or failed killing attempt) were considered regular dives. Current strength was defined on a qualitative scale as: 0 = none, 1 = weak (just noticeable, but strong enough to determine direction), 2 = medium, 3 = strong (feels difficult/uncomfortable to swim against).

On each dive, divers recorded the number of sharks observed per species, the sex of each shark, based on the presence or absence of claspers (male/female/unknown), and the estimated total length of each individual (TL, cm). In Cayman most reef-associated shark species (except *Carcharhinus* spp. and *Sphyrna* spp.) are easily distinguished, but divers were instructed to refer to the supplied training material or contact the coordinator if there were doubts about the identification of a species. Given issues associated with replicate counts of mobile predators such as sharks [87], for each dive when more than one diver was present, the final entry on the number of sharks encountered was decided in consultation with the entire dive group.

Divers recorded data on the presence/absence of any dorsal fin tag (fitted during other ongoing research [12,83]), the apparent health of the animal (e.g., scars, injuries), and any other visible distinguishing features on the body that might help subsequent identification of the particular individual. If the shark was an individual known to divers, its "pet" name (e.g., Finn) and identifying markings (e.g., scar on right side corner mouth) were reported. If the shark was unknown to divers, but possessed distinguishing markings (e.g., scars, injuries, hooks, morphometric features), a detailed description of the markings was reported (e.g., large nick in 1ˢᵗ dorsal fin and large hook in mouth) to aid subsequent individual identification.

To quantify shark behaviour, divers were encouraged to report the depth of the shark encounter and the shark's behaviour, including shark-diver, shark-shark or shark-prey interactions, and any evidence of reproductive behaviour (e.g., courting behaviour or, mating scars on females, as listed in the training material). To reduce bias, divers were asked to report data in their preferred unit (metric or imperial). Further instructions and assumptions for each variable are detailed in S1 Table.

## Data organization and analysis

The raw dive data obtained from divers were processed at both Shark Log and database levels. If data for dives were incomplete, individual divers were contacted to attempt to obtain the missing information. Dives without date or any information on location (e.g., area, dive site, or coordinates) were rejected entirely. Data from accepted dives were standardized and validated. As the popular names of some dive sites differed between divers and changed within the diving community over time, the dive locations (dive site name, coordinates) were standardized and, if necessary, missing information (e.g., coordinates) added. Temperature (°C), depth (m), visibility (m), and shark size (TL, cm), if reported as imperial, were converted into metric units, and dive duration was standardized to minutes (min). For night dives (17:00–05:59), even though Cayman's clear water may allow for up to 5–10 m visibilities on night dives (e.g., on a full moon), the visibility was standardized to 0 m.

Data on species and sex of individual sharks provided by divers were accepted, except for *Sphyrna* spp. which were analyzed collectively for 'hammerhead spp.' [45]. The number of sharks per dive were accepted, except for outliers (> 5 sharks) unless the outlier was validated by the coordinator (JK) who confirmed the number through additional communication with the relevant diver or dive group. Acknowledging the known inaccuracies with visual length estimates [88], sharks were grouped into maturity stages (immature, mature) using the raw size estimates (Table 1).

To assist the recognition of individual sharks, each shark described received an ID number, and an up-to-date record of its distinguishing markings was maintained (examples shown on Fig 3). A shark could be identified individually by matching its "pet" name or the description of its markings to the reference list (S2 Table). This was based on the assumptions (1) that most markings would not change markedly during the study period and (2) that individual

sharks were encountered frequently enough to document any changes in appearance. Particular issues included difficulty in reading the number on any dorsal fin tag and that the "pet" names given to individual sharks sometimes differed between divers. Remaining comments, including those on injuries and reproductive activities, were reviewed and then summarized for each species.

Finally, any duplicate dives were removed; duplicates originated when multiple divers dived together, and more than one diver reported the same dive. In some rare cases in which different shark counts were reported for the same dive, divers were contacted and only the

**Table 1. Species-specific total length estimates (cm) for mature sharks recorded by the Sharklogger Network in the Cayman Islands.**

| Species | Mature sharks (cm) | Reference |
|---|---|---|
| Caribbean reef (*Carcharhinus perezi*) and blacktip shark (*Carcharhinus limbatus*) | > 150 | [31,89] |
| nurse shark (*Ginglymostoma cirratum*) | > 180 | [89,90] |
| lemon shark (*Negaprion brevirostris*) | > 200 | [89] |
| hammerhead spp. (*Sphyrna* spp.) | > 200 | |
| tiger shark (*Galeocerdo cuvier*) | > 200 | |
| silky shark (*Carcharhinus falciformis*) | > 150 | [91] |
| whale shark (*Rhincodon typus*) | > 900 | [92] |

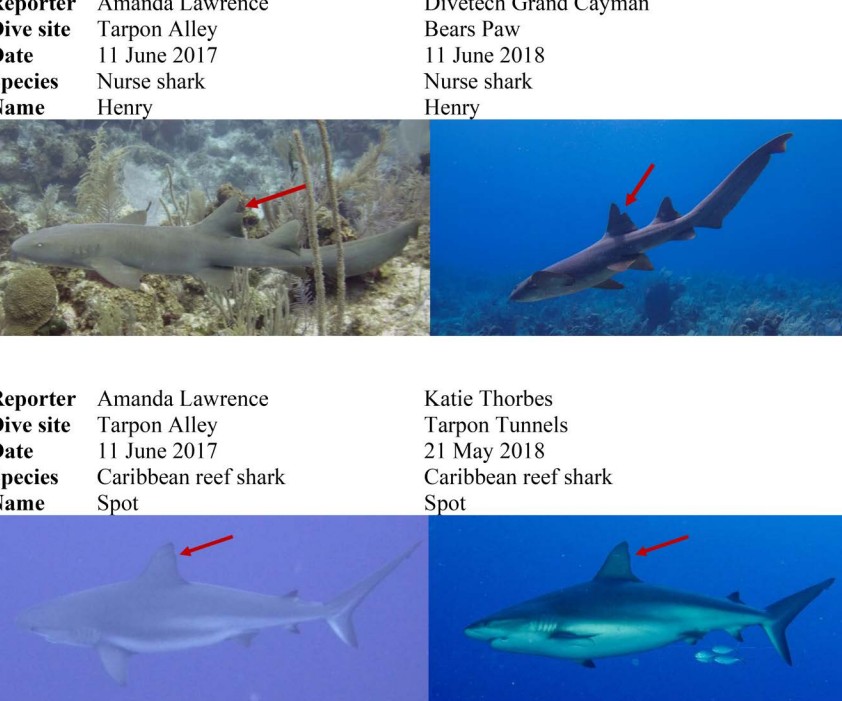

| **Reporter** | Amanda Lawrence | Divetech Grand Cayman |
|---|---|---|
| **Dive site** | Tarpon Alley | Bears Paw |
| **Date** | 11 June 2017 | 11 June 2018 |
| **Species** | Nurse shark | Nurse shark |
| **Name** | Henry | Henry |

| **Reporter** | Amanda Lawrence | Katie Thorbes |
|---|---|---|
| **Dive site** | Tarpon Alley | Tarpon Tunnels |
| **Date** | 11 June 2017 | 21 May 2018 |
| **Species** | Caribbean reef shark | Caribbean reef shark |
| **Name** | Spot | Spot |

**Fig 3. Examples of resightings of individually identified sharks.** Red arrow points towards their distinguishable markings and information about the diver, dive site, and date reported on submitted Shark Logs are included.

most reliable data retained. The processing of the data generated 24,442 unique dives over the two-year study period.

To examine the relative abundance of sharks, the shark count data were tested to determine whether assumptions of normality and homogeneity of variances were met. Even after log, square-root, and z score transformations, data were significantly different (Lilliefors (Kolmogorov–Smirnov) test: $p < 0.05$) from a normal distribution; consequently, non-parametric tests were performed on untransformed data. For each statistical analysis, homogeneity of variances was tested using a Levene's test ($\alpha = 0.05$). If the assumption of homogeneity was rejected ($p < 0.05$), the subsequent statistical test was set to assume unequal variances. Statistical test results were considered significant at the 0.05 level ($p < 0.05$), in which case the null hypothesis ($H_0$) was rejected, and the alternative hypothesis ($H_1$) accepted. For p-values greater than 0.001 the exact number was reported (e.g., $p = 0.0314$) but values smaller than 0.001 were reported as '$p < 0.001$'. Correlation of variables was given if rho $\geq 0.01$ and considered '0' (i.e., no correlation) if rho $< 0.01$. All statistical analyses were performed in R Studio (R v3.6.1) using packages 'car', 'dunn.test' and 'DescTools', maps were visualized using ESRI's ArcGIS Desktop v10.4, and graphs were drawn in Microsoft Excel (v1911).

Recognizing that there will have been variation in the detectability of sharks [93,94] and that direct measurement of detectability was not possible [95], variables that were likely to have affected the detection of sharks rather than patterns in true shark abundance (i.e., time of day, dive duration, lionfish culling, dive group size) were also assessed and reported.

**Relative abundance of sharks.** For each species, the spatial distribution was visualized by mapping the mean (sharks/dive) at dive site level using ArcGIS. The null hypotheses that the relative abundance (sharks/dive) was not significantly different between species, between islands, between areas of islands (see Fig 1 for details), or between months of the year were tested using a non-parametric two-way Kruskal-Wallis rank sum test and any significant differences were determined with a post-hoc Dunn test. To test the null hypothesis that the relative abundance (sharks/dive) of a species did not differ between years (2017/2018), between seasons (summer: April - September, winter: October - March) or between MPA/non-MPAs, a non-parametric, two-way Mann-Whitney U test was used.

To examine the demographic composition of each species, only sharks for which sex and maturity stage, respectively, were determined with confidence were included in the analysis. For each species, the numbers of each demographic were then compared using a Chi-square goodness of fit test.

**Abiotic drivers of sharks in the Cayman Islands.** A non-parametric two-way Kruskal-Wallis rank sum test was used to test the null hypothesis that the relative abundance (sharks/dive) of a species did not differ significantly between depth categories (binned in 5 m intervals) and between current strength categories (0 - 3). Subsequently a post-hoc Dunn test was used to identify potentially significant differences within each group. The relative abundance (shark/dive) was plotted against the water temperature and the relationship assessed using a non-parametric Spearman's rank correlation [96].

**Effects of dive behaviour on shark abundance.** To test the null hypothesis that the relative abundance (sharks/dive) across all species (n = 8) did not differ significantly between dives with and without lionfish culling events, a non-parametric, two-way Mann-Whitney U test was used. For each variable (dive duration (min), time of day, visibility (m), dive group size (the number of divers)), the mean (±SE) and range were reported. For each variable, the significance of the relationship between the variable and the relative abundance (sharks/dive) across all species (n = 8) was assessed using a non-parametric Spearman's rank correlation.

**Behavioral patterns of sharks in the Cayman Islands.** The size of linear home ranges and degree of site-fidelity of individually identified sharks were assessed by (1) defining the

minimum linear displacement (MLD) between the furthest apart sightings of the same shark, and (2) calculating a standardized Site-Fidelity Index (SFI) to identify the degree of site attachment (similar to ongoing local shark research [83,84].

The MLD was defined as the linear distance between the two most distant dive sites from which an individually recognized shark was reported. The position data of the two dive sites were converted into distance (km), using the Great-circle distance formula (https://www.movable-type.co.uk/scripts/latlong.html). The SFI was based on the sighting frequencies for the individual sharks at individual dive sites (Equation (1)); these were used to rank the sites where the shark was reported. The MLD and SFI for different individuals were then pooled for each species and the mean ( ± SE) and range reported.

$$\text{SFI ( \%)} = \frac{\text{number of shark sightings at dive site}}{\text{sum of individual sightings}} \times 100 \qquad (1)$$

For each species, the information on habitat, depth and inter-specific interactions were extracted and summarized. Adjectives used by divers to describe diver-shark interactions were extracted, grouped into demographics, and patterns that were recorded five or more times were reported. For each species, intraspecific interactions were extracted and, if possible, grouped into 'mating', 'pupping' and 'pregnancy'. Evidence for mating was defined as comments describing 'scars/ bite marks on body and head' or a successful or failed mating event. Comments referring to suddenly 'slimmed down sharks' and occurrence of 'baby sharks' ( <1 m) were considered evidence for pupping. Pregnancy was inferred from comments on girth of mature females (e.g., 'girthy shark', 'getting bigger', 'big belly'). To examine whether the reproductive behaviour changed throughout the year, across all dives and for each species, the percentage of reports including evidence of mating, pupping or pregnancy was plotted against months of the year. It was expected that sharks would exhibit reproductive behaviour, especially mating and pupping, in summer (April–September) and therefore the percentage of reports would increase in those months.

**Programme management.** Across the study period, the numbers of participating divers was reported. Monthly engagement was defined as the ratio of the number of submitted Shark Logs to the number of recruited divers (Equation (2)). The failure to submit a Shark Log was due either to the diver losing interest or to the lack of diving in that month. The engagement, the number of Shark Logs, number of new recruits, and number of drop-outs were plotted against months for visual interpretation and the mean ( ± SE) and range reported.

$$\text{Engagement} = \frac{\text{number of Shark Logs}}{\text{number of recruited divers}} \times 100 \qquad (2)$$

It is to be noted that sample sizes differed among variables (S3 Table) because on occasion particular information was missing from some Shark Logs.

**Ethics statement and permits.** The Cayman Islands Department of Environment (DoE) and the National Conservation Council (NCC) approved the citizen science program and its protocols and confirmed that the use of only recreational observations required no research permits. This study neither captured any sharks nor used any personal information about human participants. The dive and shark data were anonymized before use. The participation by divers and snorkelers from the Cayman community was voluntarily and every participant was aware of the use of the data for this study, automatically confirmed consent by submitting the Shark Log datasheet to the program coordinator and those who agreed to be mentioned by name are listed on the DoE website.

## Results

Over the two-year study period, 69 divers conducted 24,442 dives, reporting 4,666 shark sightings from eight shark species, across 472 dives sites. Dives were performed by at least one diver on every day of the calendar year, at times between 6:00–22:30, with a mean rate of 33.90 dives/day (± 0.51 SE; 1–98 dives/day). The mean number of dives per site was 51.57 (± 3.75 SE; 1–738 dives/dive site). The distribution of diving effort by divers during the study period can be viewed in the Supporting Information (S3 Fig).

### Species diversity and relative abundance

Divers reported a mean of 0.19 ± 0.0035 (SE) sharks/dive (0–10 sharks/dive) from six reef-associated species (Caribbean reef (*Carcharhinus perezi*), nurse (*Ginglymostoma cirratum*), hammerhead sp. (*Sphyrna* spp.), blacktip (*Carcharhinus limbatus*), tiger (*Galeocerdo cuvier*), lemon (*Negaprion brevirostris*), and two oceanic species (silky (*Carcharhinus falciformis)*, whale shark (*Rhincodon typus*)). Relative abundance was significantly different between species (Kruskal-Wallis rank sum test: $\chi^2$ = 9746, df = 7, p < 0.001) with Caribbean reef, nurse, and hammerhead spp. sharks each being significantly more abundant than the remaining species (Fig 4; post-hoc Dunn test reported in S4 Table). Due to the low numbers of the other species recorded, subsequent analyses were performed for Caribbean reef, nurse, and hammerhead spp. sharks only.

Overall, divers reported that females of each species observed were significantly more commonly than males (Chi-squared test: Caribbean reef shark: $\chi^2$ = 23.929, df = 1, p < 0.001; nurse shark: $\chi^2$ = 6.4853, df = 1, p = 0.01088) with sex ratios 1:1.3 and 1:1.4 for Caribbean reef and nurse shark respectively, while only female hammerhead spp. were reported by divers

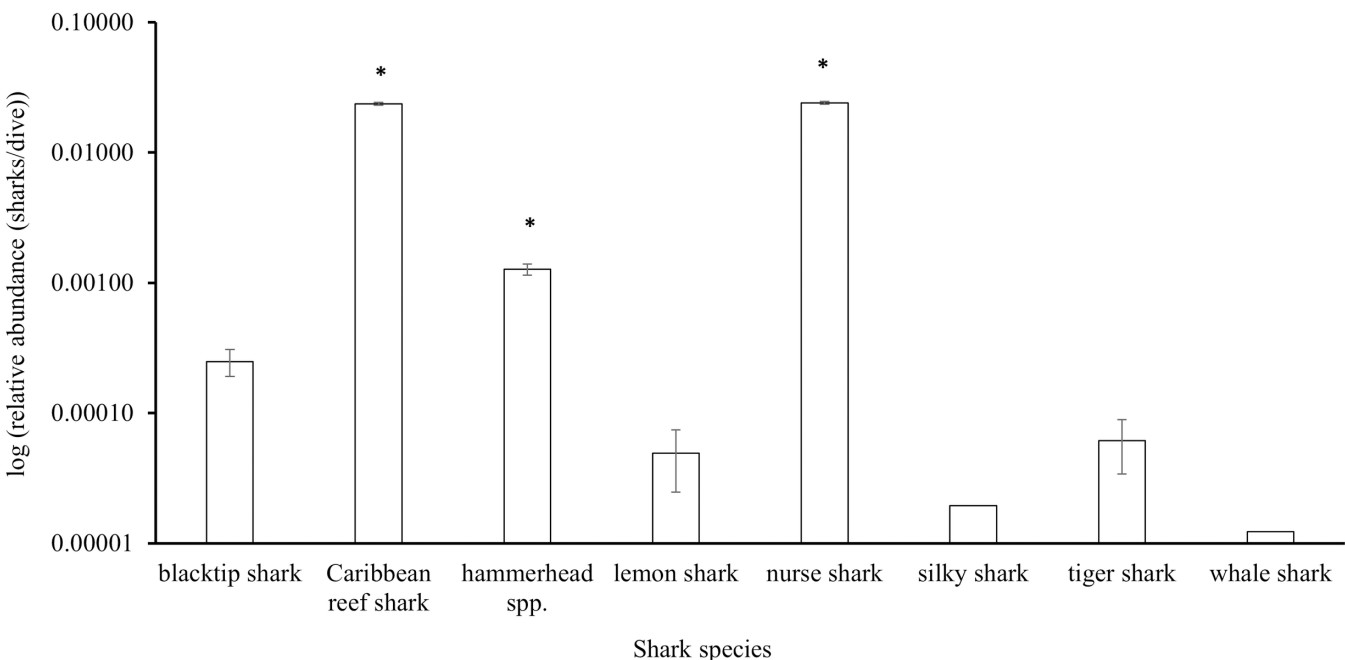

**Fig 4. Relative abundance (mean sharks/dive ± SE), plotted on a log scale, of the shark species reported by the Sharklogger Network (n = 24,442 dives) during the study period (2017 + 2018).** Species the abundance of which were significantly different (p < 0.05) from those of all other species are marked with * and post-hoc results are detailed in S4 Table.

(Table 2). Mature Caribbean reef sharks (Chi-squared test: $\chi^2$ = 300.7, df = 1, p < 0.001), hammerhead spp. (Chi-squared test: $\chi^2$ = 30.6, df = 1, p < 0.001), and immature nurse sharks (Chi-squared test: $\chi^2$ = 663.91, df = 1, p < 0.001) were significantly more common on dives than the immature stage of each species (Table 2).

## Spatial distribution of sharks

The relative abundance (sharks/dive) of all sharks combined (n = 8 species) was significantly different between islands (Kruskal–Wallis rank sum test: $\chi^2$ = 157.82, df = 2, p < 0.001) with a significantly (post-hoc Dunn results in S5 Table) greater abundance of sharks in Little Cayman followed by Grand Cayman and Cayman Brac (Fig 5). For each species taken separately, only the relative abundance of nurse sharks was significantly different between islands (Kruskal-Wallis rank sum test: $\chi^2$ = 221.06, df = 2, p < 0.001), with a greater abundance on Little Cayman (post-hoc Dunn results in S5 Table) than on the other islands. The differences in relative abundance were not significantly different for Caribbean reef sharks

**Table 2. Demographics for Caribbean reef sharks, nurse sharks, and hammerhead spp. reported by the Sharklogger Network (n = 24,4442 dives) during the study period (2017 + 2018).**

| Species | Number of sightings | | | | | | |
|---|---|---|---|---|---|---|---|
| | Total | No. with sex determined | Female sharks | Male sharks | No. with maturity determined | Immature sharks | Mature sharks |
| Caribbean reef shark | 2319 | 1,124 (48.5%) | 644 | 480 | 1,982 (85.5%) | 605 | 1,377 |
| Nurse shark | 2040 | 272 (13.3%) | 157 | 115 | 1,648 (80.8%) | 1,347 | 301 |
| Hammerhead spp. | 103 | 5 (4.9%) | 5 | 0 | 85 (82.5%) | 17 | 68 |

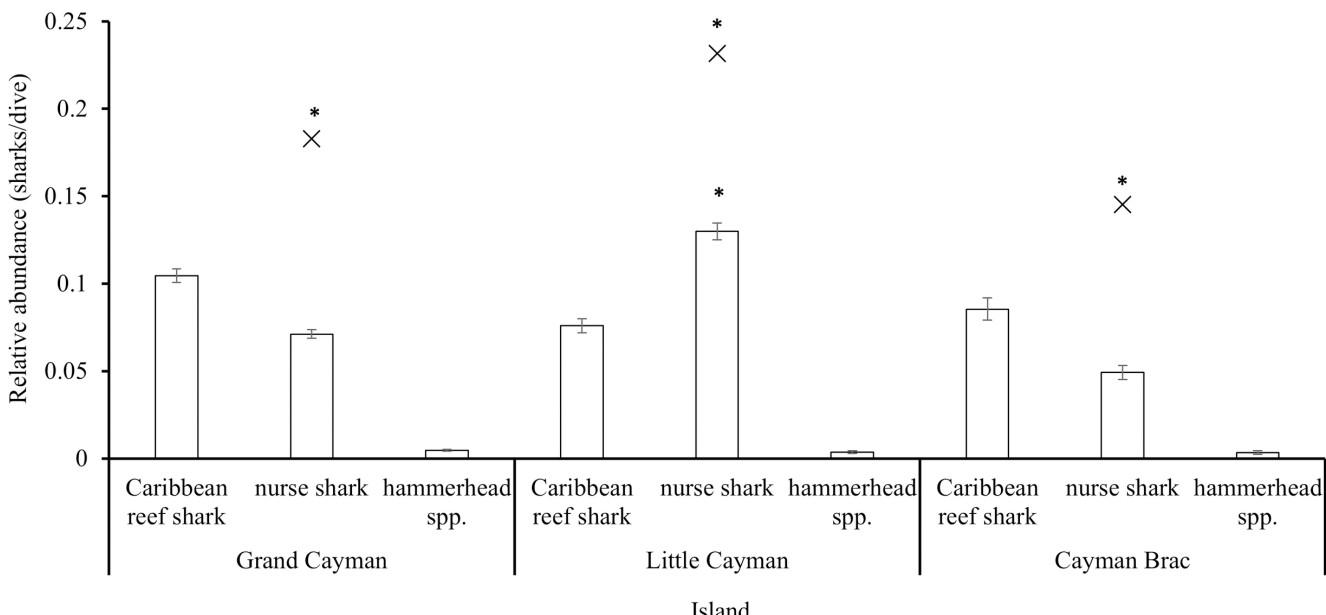

**Fig 5. Relative abundance (mean sharks/dive ± SE) of Caribbean reef shark, nurse shark and hammerhead spp. on each of the Cayman Islands reported by the Sharklogger Network (n = 24,442 dives) during the study period (2017 + 2018).** X indicates the mean for all sharks combined (n = 8 species), with significant differences from all other comparable values marked with * and post-hoc results reported in S5 Table. Species-specific abundances that are significantly different (p < 0.05) to all other abundances for that species are marked with * and remaining post-hoc results are detailed in S5 Table.

(Kruskal-Wallis rank sum test: $\chi^2$ = 0.17041, df = 2, p = 0.9183) or hammerhead spp. $\chi^2$ = 1.8158, df = 2, p = 0.4034, Fig 5).

The relative abundance (sharks/dive) of Caribbean reef sharks varied significantly between different areas on Grand Cayman (Kruskal-Wallis rank sum test: $\chi^2$ = 1511.4, df = 6, p < 0.001), Little Cayman (Kruskal–Wallis rank sum test: $\chi^2$ = 49.134, df = 3, p < 0.001) and Cayman Brac (Kruskal–Wallis rank sum test: $\chi^2$ = 254.74, df = 3, p < 0.001); no Caribbean reef sharks were reported in two sectors on Grand Cayman (SW, W) nor two sectors on Cayman Brac (E, N), nor in one sector (S) on Little Cayman (Fig 6). Caribbean reef sharks were

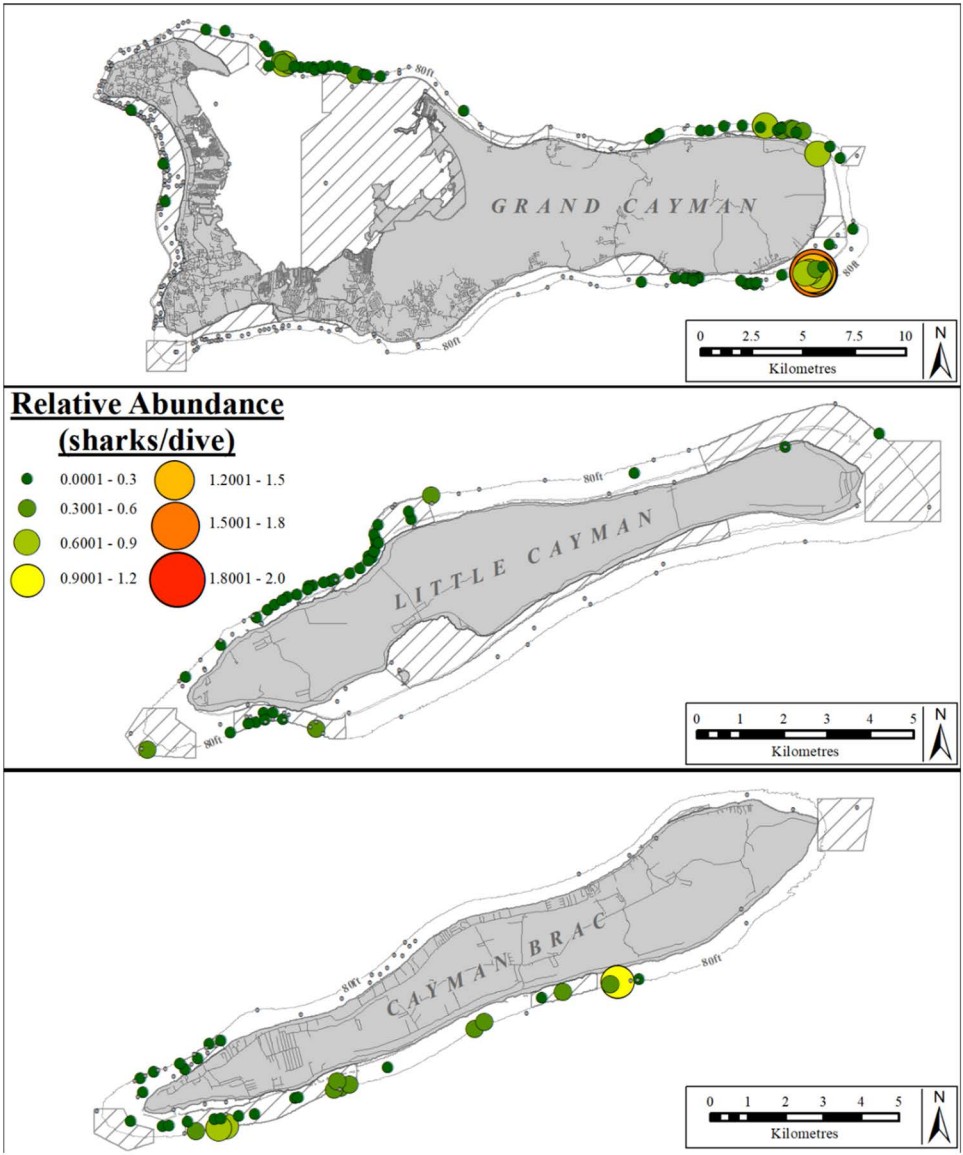

**Fig 6. Relative abundance (sharks/dive) of Caribbean reef sharks at dive sites reported by the Sharklogger Network (n = 24,442 dives) during the study period (2017 + 2018).** Grey dots (in case of no sharks) or color-coded circles (in case of shark sightings) indicate dive sites, with circle size and color both scaled to indicate relative abundance (i.e., dive sites with more sharks/dive are indicted by both a larger circles and a different color, compared to dive sites with fewer sharks/dive); color spectrum ranges from few sharks/dive (dark green) to most sharks/dive (red). Relative abundances ranged from 0.001–1.67 sharks/dive.

significantly more abundant in the E, NE, and NW sectors of Grand Cayman (post-hoc Dunn results S6 Table), in the N sector of Little Cayman (post-hoc Dunn results S7 Table), and the S sector of Cayman Brac (post-hoc Dunn results S7 Table) compared to the remaining areas of each island, respectively.

Nurse sharks were sighted in all areas across the Cayman Islands and their mean abundance (sharks/dive) was only significantly different between different sectors on Grand Cayman (Kruskal–Wallis rank sum test: $\chi^2$ = 214.12, df = 6, p < 0.001) and on Little Cayman (Kruskal-Wallis rank sum test: $\chi^2$ = 25.616, df = 3, p < 0.001; Fig 7), but not between sectors

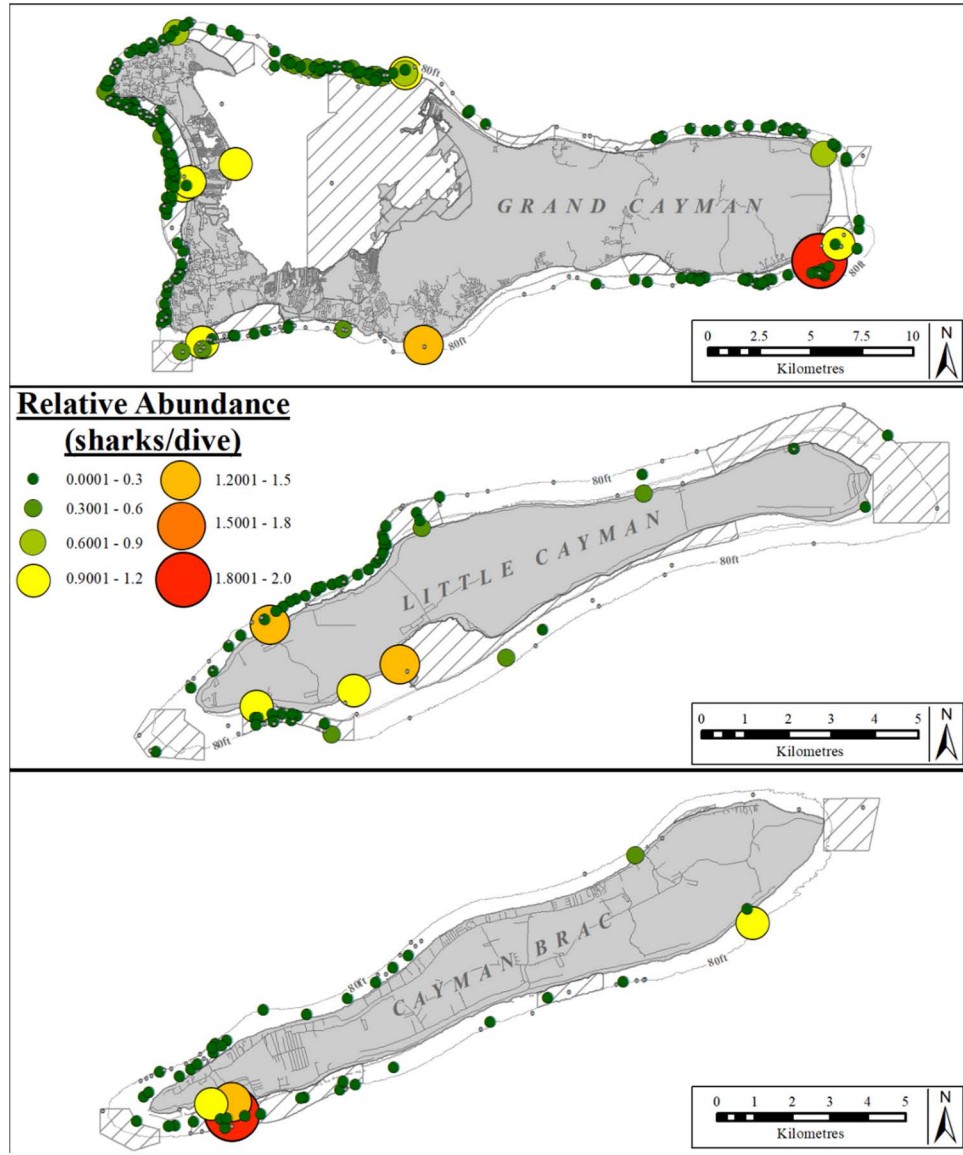

**Fig 7. Relative abundance (sharks/dive) of nurse sharks at dive sites reported by the Sharklogger Network (n = 24,442 dives) during the study period (2017 + 2018).** Grey dots (in case of no sharks) or color-coded circles (in case of shark sightings) indicate dive sites, with circle size and color both scaled to indicate relative abundance (i.e., dive sites with more sharks/dive are indicted by both a larger circles and a different color, compared to dive sites with fewer sharks/dive); color spectrum ranges from few sharks/dive (dark green) to most sharks/dive (red). Relative abundance ranged from 0.001 - 2 sharks/dive.

on Cayman Brac (Kruskal-Wallis rank sum test: $\chi^2$ = 5.9695, df = 3, p = 0.1131). The relative abundance was significantly greater in the NW sector of Grand Cayman (post-hoc Dunn results S6 Table) than in the remaining sectors; and in the N sector compared to the E and W sectors of Little Cayman (post-hoc Dunn results S7 Table).

The relative abundance (sharks/dive) of hammerhead spp. differed significantly between sectors on Grand Cayman (Kruskal-Wallis rank sum test: $\chi^2$ = 54.851, df = 6, p < 0.001) and on Little Cayman (Kruskal-Wallis rank sum test: $\chi^2$ = 29.101, df = 3, p < 0.001), but not between sectors on Cayman Brac (Kruskal-Wallis rank sum test: $\chi^2$ = 4.4461, df = 3, p =

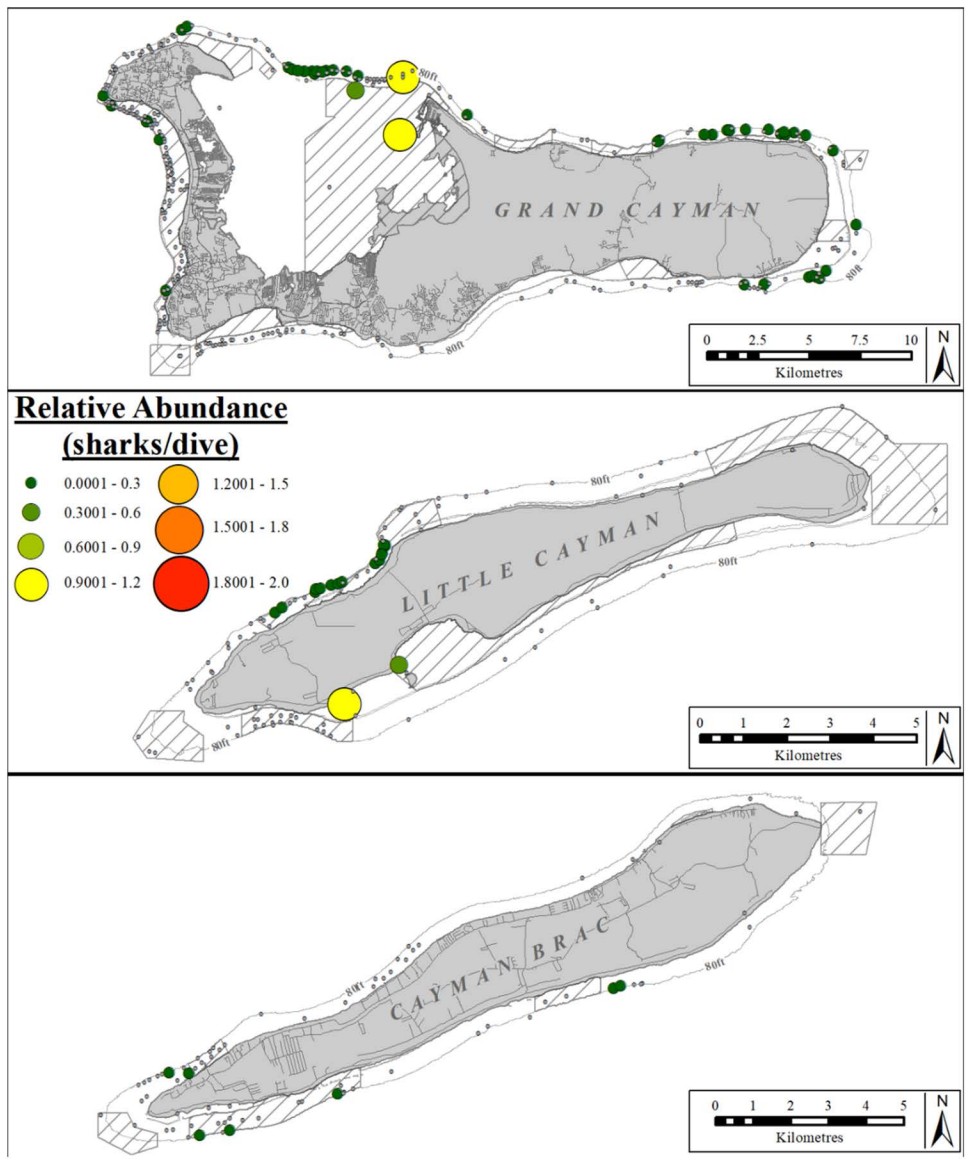

**Fig 8. Relative abundance (sharks/dive) of hammerhead spp. at dive sites reported by the Sharklogger Network (n = 24,442 dives) during the study period (2017 + 2018).** Grey dots (in case of no sharks) or color-coded circles (in case of shark sightings) indicate dive sites, with circle size and color both scaled to indicate relative abundance (i.e., dive sites with more sharks/dive are indicted by both a larger circles and a different color, compared to dive sites with fewer sharks/dive); color spectrum ranges from few sharks/dive (dark green) to most sharks/dive (red). Relative abundance ranged from 0.001–1 sharks/dive.

0.2171). There were no reports in two sectors on Grand Cayman (SW, W), two on Little Cayman (E, W), and two on Cayman Brac (E, N; Fig 8). Hammerhead spp. were most abundant in the S sector of Little Cayman (post-hoc Dunn results S7 Table) and predominantly reported in the E, NE and NS sectors of Grand Cayman. However those sectors were not significantly different from each other nor all remaining sectors (post-hoc Dunn results S6 Table).

Only nurse sharks were significantly more abundant (Mann-Whitney U test: W = 75492448, p < 0.001) inside MPAs than in non-MPA areas. Caribbean reef sharks (Mann-Whitney U test: W = 69659488, p < 0.001) and hammerhead spp. (Mann-Whitney U test: W = 73642079, p < 0.001) were significantly more abundant outside MPAs than inside them (Fig 9).

## Temporal patterns in abundance of sharks

Comparing 2017 with 2018, relative abundance was significantly less in 2018 (Mann–Whitney U test: W = 73502384, p < 0.001) for Caribbean reef shark, but significantly greater for nurse shark (Mann–Whitney U test: W = 71903719, p = 0.01775) and hammerhead spp. (Mann–Whitney U test: W = 72337254, p = 0.01096; Fig 10).

There was evidence of seasonality in Caribbean reef shark occurrence, with significantly more (Mann-Whitney U test: W = 72969807, p = 0.001303) individuals being reported by divers in summer (April–September) than during winter (October – March), but there was no significant difference in relative abundances between seasons for nurse sharks (Mann-Whitney U test: W = 74094765, p = 0.122) nor for hammerhead spp. (Mann-Whitney U test: W = 73752000, p = 0.4701, Fig 11).

Relative abundances (sharks/dive) varied significantly across months for each species (Kruskal-Wallis rank sum test: Caribbean reef shark: $\chi^2$ = 39.148, df = 11, p < 0.001; nurse shark: $\chi^2$ = 47.343, df = 11, p < 0.001; hammerhead spp.: $\chi^2$ = 33.351, df = 11, p < 0.001; Fig 12) with significantly fewer Caribbean reef sharks (post-hoc Dunn results S8 Table) in May and June compared to the other months, significantly more nurse sharks (post-hoc Dunn results S8 Table) in August compared to most other months (except November, September) and fewer in January (except compared to March; post-hoc Dunn results S8 Table), and significantly more hammerhead spp. (except compared to February; post-hoc Dunn results S8 Table) in May.

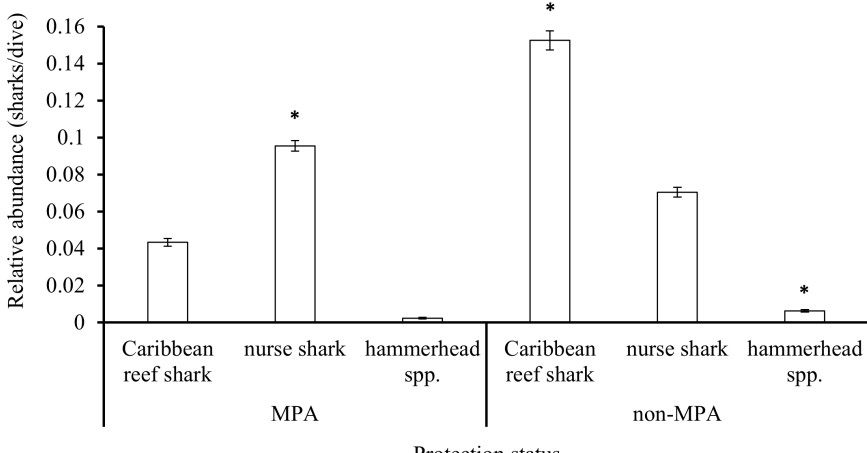

**Fig 9. Relative abundance (mean (sharks/dive) ± SE) for the three shark species inside MPA's and non-MPA's reported by the Sharklogger Network (n = 24,442 dives) during the study period (2017 + 2018).** Significant differences (p < 0.05) in relative abundances between MPA and non-MPA areas are marked with *.

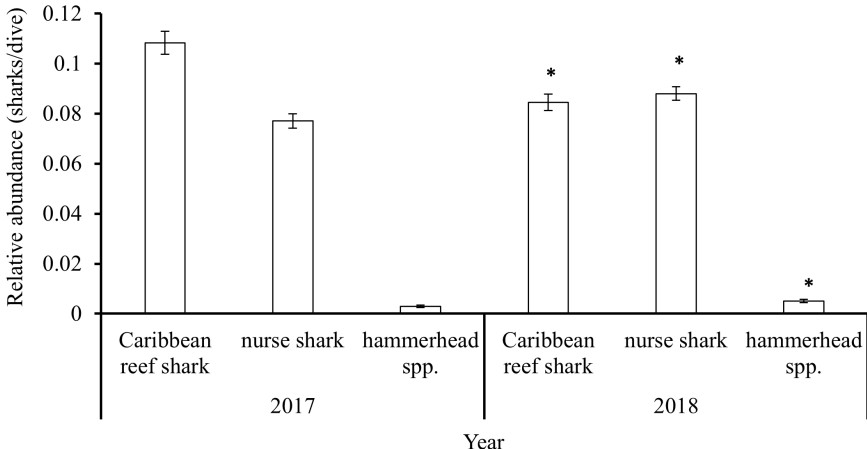

**Fig 10. Relative abundance (mean (sharks/dive) ± SE) of the three shark species across the study period (2017 + 2018) reported by the Sharklogger Network (n = 24,442 dives).** Significant differences in abundance of a given species between years are marked with *.

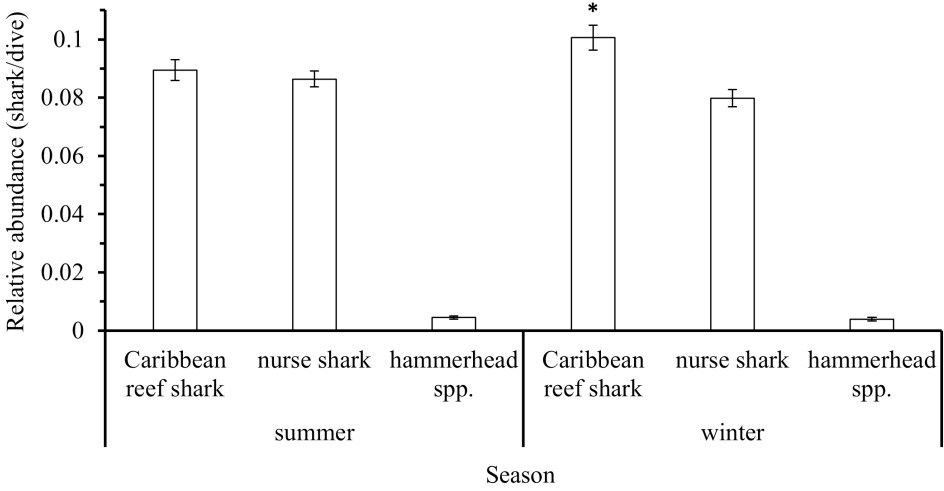

**Fig 11. Relative abundance (mean (sharks/dive) ± SE) of three shark species between seasons (summer, winter) reported by the Sharklogger Network (n = 24,442 dives) during the study period (2017 + 2018).** Significant differences in abundance between seasons are marked with *.

## Abiotic drivers of shark abundance

The mean maximum depth of all dives was 21.32 m ( ± 0.05 SE; 0–78 m). The relative abundance (sharks/dive) of each species differed significantly between depth categories (Kruskal–Wallis rank sum test: Caribbean reef shark: $\chi^2$ = 768.64, df = 9, p < 0.001; nurse shark: $\chi^2$ = 239.17, df = 9, p < 0.001; hammerhead spp.: $\chi^2$ = 78.151, df = 9, p < 0.001; Fig 13). Caribbean reef sharks appeared to be more abundant (post-hoc Dunn results S9 Table) on dives to greater than 20 m and were not reported in less than 5 m depth, nurse sharks were sighted at every depth and were significantly more abundant (post-hoc Dunn results S9 Table) in shallow water (during snorkels, represented as "0 m depth") and at 36–40 m depth,

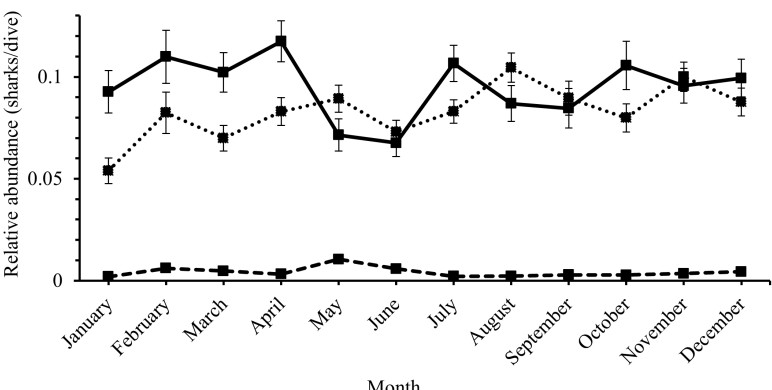

**Fig 12. Monthly variation in relative abundance (mean (sharks/dive) ± SE) of the three shark species reported by the Sharklogger Network (n = 24,442 dives) during the study period (2017 + 2018).** Caribbean reef shark = solid black line, nurse shark = dotted line, hammerhead spp. = dashed line. For each species, post-hoc results between months are detailed in S8 Table.

and hammerhead spp. were significantly more abundant (post-hoc Dunn results S9 Table) at depths greater than 40 m and absent on dives at 6–15 m depth (Fig 13).

Current strength during dives ranged from no current (0) to very strong current (3) (mean ± SE = 0.18 ± 0.003) and relative abundance (sharks/dive) of sharks (n = 8 species) varied significantly with current strength (Kruskal-Wallis rank sum test: $\chi^2$ = 189.6, df = 3, p < 0.001, Fig 14) with significantly more (post-hoc Dunn results S10 Table) sharks on dives with weak (1) to moderate (2) current strength, compared to no current (0) being present.

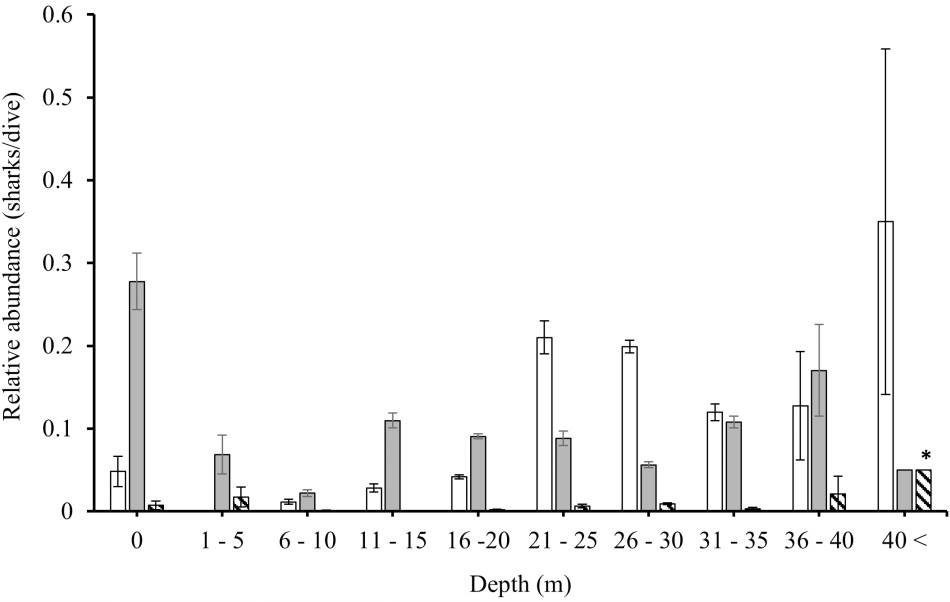

**Fig 13. Relative abundance (mean (sharks/dive) ± SE) of species at depth categories reported by the Sharklogger Network (n = 22,703 dives) during the study period (2017 + 2018).** Caribbean reef shark = white, nurse shark = grey, and hammerhead spp. = hatched bar. Species-specific abundances that are significantly different (p < 0.05) to all other depth categories for that species are marked with * and remaining post-hoc results are detailed in S9 Table.

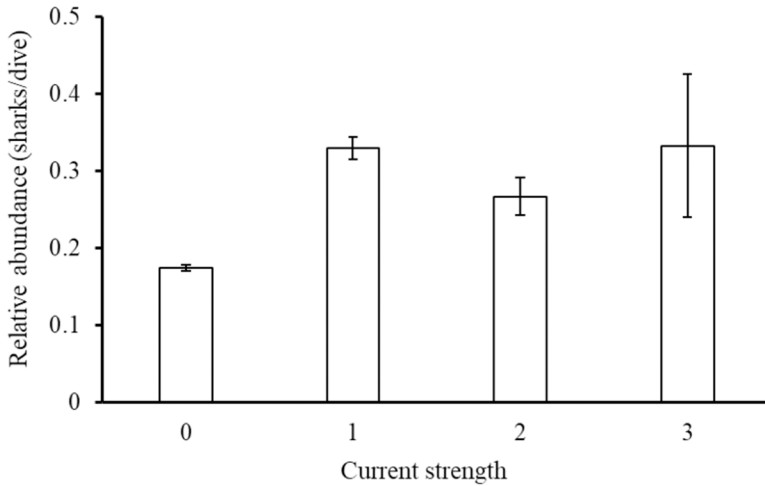

**Fig 14. Relative shark abundance (mean (sharks/dive) ± SE) from all shark species (n = 8) combined against current strength reported by the Sharklogger Network (n = 22,703 dives) in the Cayman Islands during the study period (2017 + 2018).** Significant differences between current strengths are detailed in S10 Table.

The ambient water temperature (hereafter called 'temperature') on dives ranged from 21–35 °C (mean ± SE = 27.88 ± 0.008 °C), but there was no correlation between temperature and relative abundances (sharks/dive) of any shark species (Spearman rank correlation: Caribbean reef shark: S = $1.4964e^{+12}$, rho = -0.01887666, p = 0.006668; nurse shark: S = $1.4395e^{+12}$, rho = 0.01987646, p = 0.00428; hammerhead spp.: S = $1.4893e^{+12}$, rho = -0.01407576, p = 0.04308).

## Effects of diver behaviour on shark abundance

Lionfish culling occurred on 1,286 dives (5.6% of total dives) during which 338 shark sightings (7.8% of total shark sightings) were recorded. The relative abundance (sharks/dive) of sharks (n = 8 species) was significantly greater (Mann-Whitney U test: W = 14114631, p < 0.001, Fig 15) on dives with lionfish culling compared to dives without. Sharks (n = 8 species) occurred at any time during a dive (author, pers. obs.). Relative abundance of all shark species (n = 8) combined was not correlated with dive duration, time of day, visibility, or dive group size and are reported in Table 3.

## Behavioral patterns of sharks

The recognition of individual sharks using markings on the body was possible for 11 Caribbean reef and 16 nurse sharks (Table 4), but not for any hammerhead spp. as this species was not frequently encountered and individuals did not come close enough to divers to allow them to recognize distinct features. For each individual the minimum linear displacement (MLD) and site-fidelity Index (SFI) are reported in S11 Table and summarized for each species in Table 4. The home ranges of nurse sharks appeared to be 1.2 larger than those of Caribbean reef sharks (Table 4), with individuals of each species being sighted at up to 10 and 25 different dive sites, respectively. Some individuals of both species appeared highly site attached with 100% of the sightings of those individuals occurring at only one dive site. Site fidelity appeared to be slightly greater for nurse sharks than for Caribbean reef sharks (Table 4).

Descriptive observations on the behaviour of Caribbean reef and nurse sharks are summarized in S12 Table. Overall, Caribbean reef sharks were described as more active than nurse sharks, with all Caribbean reef sharks being encounter while they were swimming over the reef or along the drop off at a greater depth (mean ± SE = 25.52 m ± 0.16), usually below the divers (i.e., > 30 m which is the maximum depth limit for recreational divers). Caribbean reef sharks were described as inquisitive yet cautious of divers, although individuals were less cautious (i.e., divers encountered sharks at shorter distance away) at sites which are frequently used by anglers or close to frequent fishing sites. Most nurse sharks (66%) were encountered while they were resting on the bottom either fully or partially covered by some form of ledge (e.g., crevice in coral reef, rock cave) on the shallow reef (mean ± SE = 11.25 m ± 0.4); they were rarely encountered swimming along the reef edge or the drop off (max. 37.5 m depth).

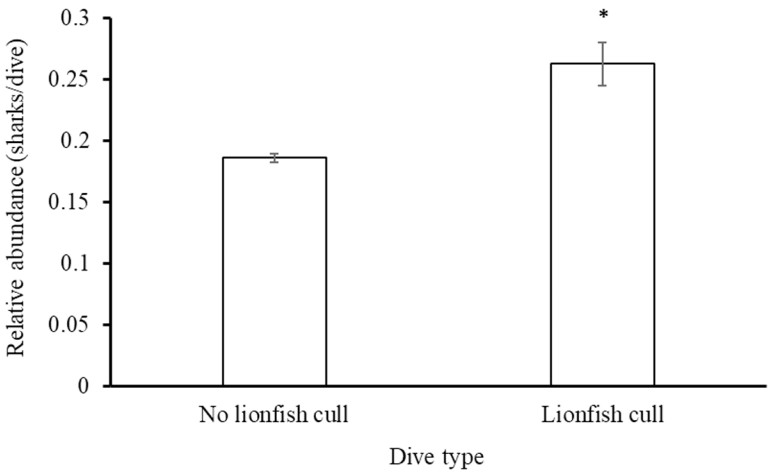

**Fig 15. Relative abundance (mean (sharks/dive) ± SE) on dives (n = 24,442) with and without lionfish culling reported by the Sharklogger Network during the study period (2017 + 2018).** Significant difference (p < 0.05) is marked with *.

**Table 3. Summary of mean (± SE) dive duration, time of day, visibility, and dive group size and statistics for the Spearman rank correlation between each variable and relative abundance (sharks/dive) of sharks (n = 8 species) reported by the Sharklogger Network during the study period (2017 + 2018).**

| Variable | Mean ± SE | Range | Spearman rank correlation |
|---|---|---|---|
| Dive duration | 50 min ± 0.084 | 5 - 200 min | S = 2.4116e + 12, rho = 0.007, p = 0.266 |
| Time of day | N/A | 6:00–22:30 | S = 2.5543e + 12, rho = -0.050, p = 0.001 |
| Visibility | 25.88 m ± 0.04 | 0–60 m | S = 1.0879e + 12, rho = 0.021, p = 0.003 |
| Dive group size | 10.22 divers ± 0.03 | 0–100 divers | S = 8.0555e + 11, rho = 0.036, p < 0.001 |

**Table 4. Summary of results for identified Caribbean reef and nurse sharks. The number of individual sharks, number of sightings, mean number of dive sites, home range measured via the minimum linear displacement (MLD), and site-attachment measured via the Site Fidelity Index (SFI) derived from dives (n = 24,442) reported by the Sharklogger Network in the Cayman Islands during the study period (2017 + 2018).**

| Species | Caribbean reef shark | nurse shark |
|---|---|---|
| Number of identified sharks | 11 | 16 |
| Number of sightings | 256 | 188 |
| Mean number of dive sites | 5.09 ± 0.73 (SE) | 5.06 ± 1.67 (SE) |
| Mean MLD (km) | 2.54 ± 0.90 (SE) | 3.29 ± 0.8 (SE) |
| Mean SFI (%) | 49.72 ± 6.68 (SE) | 52.7 ± 7.28 (SE) |

Generally, nurse sharks were less cautious than Caribbean reef sharks on encountering divers; observers often described them as 'friendly'. Sharks of both species could be encountered at any time during the dive, but, depending on the individual shark concerned, could stay with the divers for the remainder of the dive. On some consecutive dives, the shark, after joining the divers on their first dive, reappeared at the second separate dive site after the boat had moved (the sites usually being a few hundred meters apart). Reports of interspecific behaviour included interactions between the two shark species, of predation by nurse sharks, and of non-agonistic interactions of Caribbean reef sharks with groupers (*Epinephelus* spp. and *Mycteroperca* spp.), and jacks (*Caranx* spp.). Intraspecific interactions described suggest some degree of size dominance in both species while 'schooling' behaviour was reported for nurse sharks only.

There was evidence of reproductive behaviour in both common species (S12 Table). On one occasion two Caribbean reef sharks were observed mating (August, verified by JK through personal communication with the diver), and there were increased reports suggestive of mating activities from June to September, peaking in August (Fig 16). 'Pupping behaviour' of Caribbean reef sharks was also reported over summer, with a steady increase from May to August and a peak in October (Fig 16). Females described as 'pregnant' were sighted from January to October, with most reports being in March and June (Fig 16). For Caribbean reef sharks, most reports of reproductive behaviour were from the north side of Grand Cayman (E, NW) and of Little Cayman (N), and there was no evidence from Cayman Brac.

Nurse sharks were never observed mating nor was behaviour suggestive of mating activities reported, but evidence of 'pupping' in the form of very small (< 1 m) sharks was reported all year around (January - December), from all three islands (S12 Table), with most reports being from Little Cayman (N) in April and July (Fig 17). There was only one report of a female described as 'pregnant' which was from Cayman Brac (W) in June.

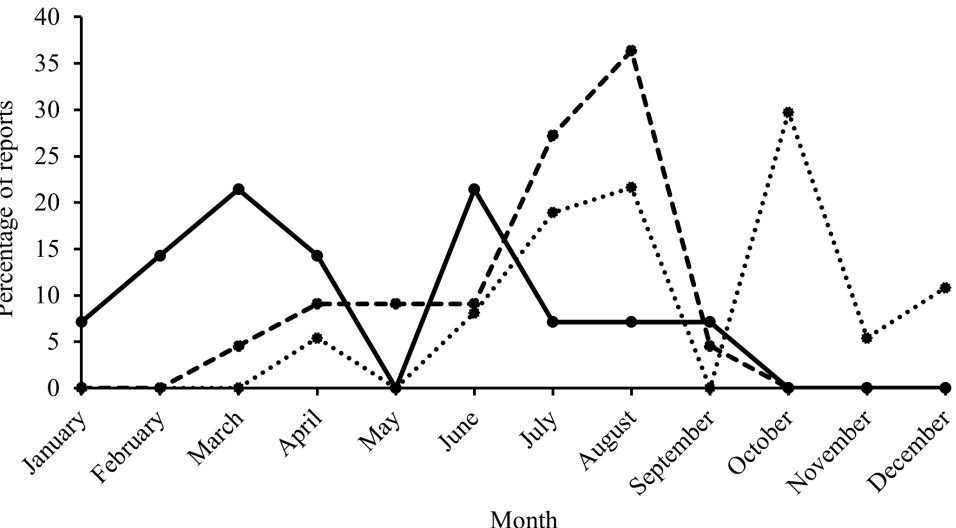

**Fig 16. Percentage of reports, describing different types of reproductive behaviour for Caribbean reef sharks across months reported by the Sharklogger Network (n = 24,442 dives) in the Cayman Islands during the study period (2017 + 2018).** Pregnancy = solid black line, mating = dashed line, pupping = dotted line.

## Programme management

One of the authors (JK) approached a total of 125 contacts (individuals and dive centers) with the objective to recruit volunteers over the two-year study period. A total of 64 divers (50.4%) were recruited while the remaining 61 contacts were interested, but considered recruitment failures since they never submitted a Shark Log. The monthly number of recruited divers, Shark Logs, new recruits, and dropouts are summarized in Table 5; the mean level of engagement per month (i.e., the percentage of Shark Logs returned) was 77.49% (± 2.35 SE) and never fell below 50% (range = 52.78–100%).

In each year, the monthly "engagement" of volunteers appeared to be highest from January to August and to decrease in September 2017 and October 2018 (Fig 18). Despite the varying engagement, the actual number of Shark Logs returned each month increased over the first 1.5 years (January 2017–June 2018) but decreased thereafter (Fig 18).

Over the study period, new divers were recruited (= new recruits) in most months, but no new recruits occurred in four months (January, June and December of 2017 and September 2018; Fig 19). Dropouts occurred in 19 months, between March 2017– December 2018, but never exceed number of new recruits during the study period (Fig 19).

## Discussion

The multiple benefits of citizen science are widely recognized; nevertheless this study represents one of the few volunteer-based programs to have collected effort-based data on shark

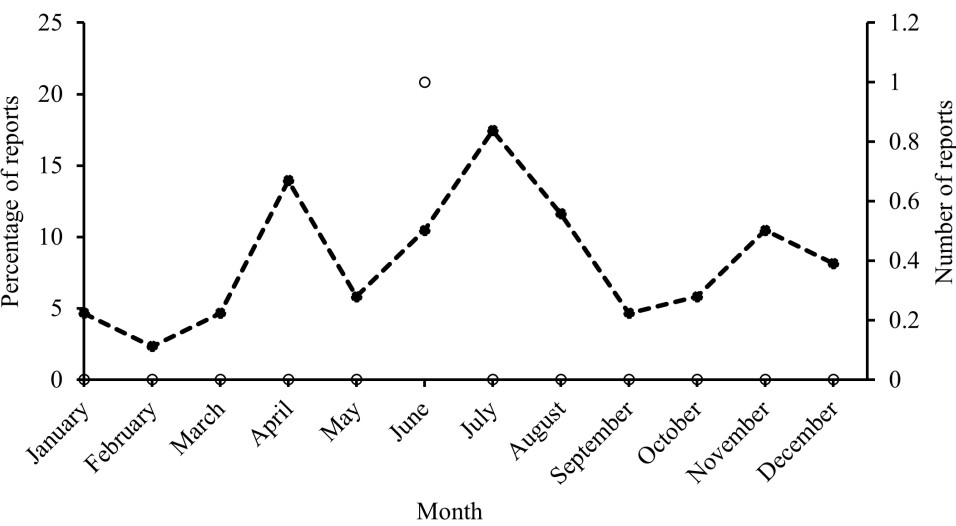

**Fig 17. Variation in numbers of reports of pupping-related behaviour (as percentage of all report, dashed line) and a report of pregnancy (as actual number, black circle) in nurse sharks across months reported by the Sharklogger Network (n = 24,442 dives) during the study period (2017 + 2018).**

**Table 5. Summary of monthly recruitment and management results from the Sharklogger Network over the study period (2017 + 2018).**

| Number of | Mean ± SE | Range |
|---|---|---|
| recruited divers participated per month | 27.25 ± 1.52 | 12 - 36 |
| Shark Logs per month | 20.58 ± 0.98 | 12 - 29 |
| new recruits per month | 2 ± 0.28 | 0 - 5 |
| drop-outs per month | 1.08 ± 0.19 | 0 - 3 |

abundance (see [54,77,78,97]) especially in such quantity. The data demonstrated that when all dives were recorded by participants, irrespective of whether sharks were present or not, such a program can be invaluable for monitoring populations of coastal sharks. The combined survey effort by the Sharklogger Network in the Cayman Islands was considerable and given the large number of reports found to be sufficient to detect even small differences in relative abundance and in behaviour between species, islands, areas, and time of year. The analyses of data from the dive logs provided evidence for: (1) the presence of eight shark species in the Cayman Islands, with Caribbean reef shark, nurse shark and hammerhead spp. being the most abundant and present throughout the year, (2) a greater relative abundance of sharks on Little Cayman than on Grand Cayman, with fewer still on Cayman Brac, (3)

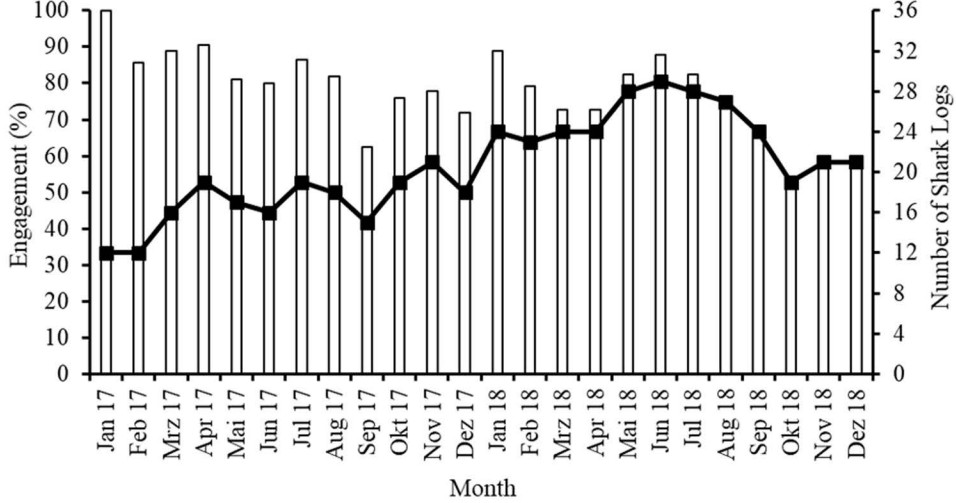

**Fig 18. Monthly number of Shark Logs returned (line) and percentage engagement (percentage of divers from the Sharklogger Network returning logs) (bars) over the study period.**

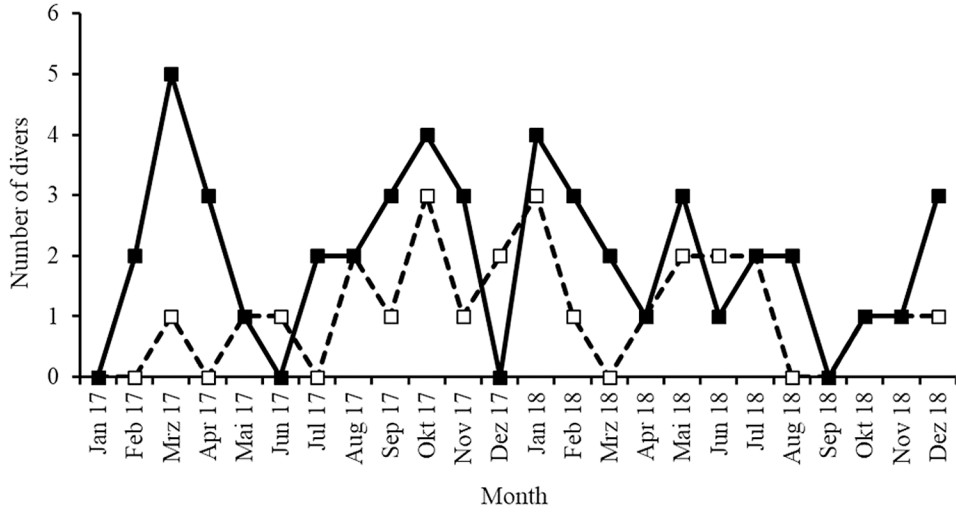

**Fig 19. Monthly number of divers classified as new recruits (solid line) and drop-outs (dotted line) from the Sharklogger Network over the study period.**

greater relative abundances of sharks in areas with less anthropogenic activity and greater exposure to strong currents, regardless of protected area status (MPA), (4) variation in the seasonal abundance of Caribbean reef sharks, with more sightings in winter, (5) evidence of reproductive behaviour occurring predominantly between May and August, (6) species-specific depth and habitat preferences, (7) site-fidelity and home range sizes in both Caribbean reef and nurse sharks.

## Effects of diver behaviour on shark abundance and data quality

As with any survey, the efficiency of the Sharklogger Network in assessing overall abundance and general behaviour will depend on the extent of overlap between the surveyed areas and the distribution of the target species, in this case sharks. Particularly in Little Cayman effort was not evenly distributed, since most diving was conducted either in the Bloody Bay Marine Park on the NW side or in the Preston Bay Marine Park on the SW side, while the remaining areas were rarely if ever dived (S3 Fig). On the other two islands the spatial distribution of dives was relatively even (S3 Fig) and, given the large dataset generated in this study, it seems reasonable to draw general conclusions about the spatial distribution of sharks in Cayman.

It is acknowledged that data collected by SCUBA divers will be subject to some degree of error arising from variations in dive conditions, e.g., visibility (also observed by Van Beek et al. [98]) and from potential behavioral changes of individual sharks when confronted with SCUBA divers (also observed by Bradley et al. [68], MacNeil et al. [69], and Cubero-Pardo et al. [99]). In this present study, different aspects of dive circumstances (time of day, visibility, dive group size, dive duration) had no significant effect on the recorded abundance of sharks. This supports findings from Palau where likewise visibility and number of divers in the water did not influence shark counts [100]. In Cayman the area sampled by divers likely remained relatively constant due to the consistently clear waters (mean visibility: 25 m) and similar routines of divers at dive sites throughout the year. An increased number of shark sightings were recorded on dives when culling of lionfish occurred, but species composition seemed to be unaffected. The scent of fish blood from dead or injured lionfish in the water, the detection of injured or struggling fish in culling devices and the sound of the spear hitting the reef during culling might have attracted additional sharks into the diver's field of view. However, culling occurred only on 5.6% of dives and therefore it seemed reasonable to use the entire data in other analyses.

Citizen science can present challenges in data quality such as potential sampling bias and participant variability (highlighted by Hyder et al. [67] and Dickinson et al. [101]). However, while dive protocols were not standardized as in a typical scientific Underwater Visual Census (UVC) (e.g., there was no defined field of view and dive time), the sampling procedures were nevertheless consistent throughout the study. The appropriate reporting of data was ensured through the standardized reporting sheet (Shark Log), and data loss was limited by insisting on monthly submissions of dive logs. Moreover, the sample protocol was kept simple, focusing only on shark and dive data. As sharks are relatively large animals, charismatic, easily detected (if within the field of view of divers), and considered 'the highlight of the dive' among most divers in Cayman and elsewhere [59], it seems likely that the simplicity of the task reduced the risk of false and non-detection events and kept the motivation of divers high throughout the study period (as emphasized by Hermoso et al. [52]). Lastly, in Cayman the majority of the diving community has an interest in shark conservation and is generally supportive of local conservation efforts. This is beneficial because the sustained motivation of stakeholders and participants has been shown to be crucial for the success of citizen science programs [77,78,97].

Although no selection of participants was made with regards to occupation, gender, age, or education level of divers or dive experience (similar to other citizen science programs, e.g., REEF https://www.reef.org/programs/volunteer-fish-survey-project), the majority of divers had advanced to expert diving experience and were considered to have local knowledge (i.e., living on Cayman, no tourists) of the study area and familiarity with target species which was assumed compensate for differences in experience [102] and reduce misidentification of species [103,104]. The close guidance and training of divers throughout the study period attempted to keep data quality constant and minimize observer error, as advised by Roelfsema et al. [105]. For example, divers were encouraged to ask questions throughout the study period and focus was given to provide detailed feedback (e.g., including pictures) within a sensible time.

While the issues in obtaining count data during non-instantaneous visual surveys are recognized [87], the risk of double counting individual sharks was considered relatively low, given that during dives in Cayman it is generally clear whether a second sighting is of a different individual shark [84,106] and because the number of sharks recorded was generally low (1 – 2 sharks). To reduce bias of length estimations, as observed by Harvey et al. [107], divers were asked to report data in their preferred metric or imperial unit because it was assumed that data were more accurate when divers reported data in familiar units (typically North Americans reported imperial and Europeans reported metric units). Nevertheless, it is likely that length estimations were biased, since other work on the accuracy of UVC estimates indicates that less experienced divers tend to overestimate the size of smaller fish, but underestimate the size of very large ones (as observed by Harvey et al. [107] and Sherman et al. [108]). In the present study it appeared that TL estimations of sharks ranged from relatively accurate to up to 30 - 40 cm in error, with most divers underestimating the size of individuals. Therefore, instead of raw TL estimates maturity stages were used for analyses.

## Species diversity

The diversity of species observed in this study is representative of coastal sharks in the wider Caribbean [12,31,34,37,45,109,110] and similar to that recorded by BRUVS (baited remote underwater video systems) in Cayman [12,30,84,106]. Several other species of sharks, not recorded by divers in this study, are known from Cayman waters [6,28,29]. These may not have been present on dives because of habitat and depth preferences outside of recreational dive limits.

Divers encountered one whale shark (during beginning of dive on the surface) and two silky sharks (JK per. obs.), both of which are less reef-associated than the remaining species reported in this study. Although silky sharks occur near reefs elsewhere [75] and whale sharks are commonly observed in some other Caribbean regions (for example: Cuba/Mexico [111], Belize [31,62], Mexican Caribbean/Honduras/Belize [112], Jamaica/Belize/Turks and Caicos Islands/Bahamas [24], Columbia [113]), both species are rarely seen on reefs in Cayman. In Cayman, silky sharks were caught on scientific longlines not far from the coast [12] and are encountered by anglers, suggesting that, locally, this species inhabits offshore waters (JK pers. obs.). Generally, there are fewer reports of whale sharks with 1-2 sightings reported by boaters each year (JK pers. obs.).

While we acknowledge some degree of bias in this study as it is natural with most citizen science data, divers in this study were encouraged to support their shark sightings with pictures or videos. However, most dives were conducted by dive centers and their safety protocols did not allow staff to carry cameras during dives. To minimize potential misidentifications, any reports of species that were questionable or where divers were unsure were verified

by the coordinator through personal communication and, if needed, referencing pictures. That the identification of species without verification by camera footage can produce meaningful results is also shown by other citizen science programs (e.g., REEF program, [46]).

The number of species recorded by divers in this study was less than reported for coastal sharks in some other Caribbean areas [31,98,109,113]. The lack of bull (*Carcharhinus leucas*), blacknose (*Carcharhinus acronotus*), and Galapagos sharks (*Carcharhinus galapagensis*) on dives in this study was expected because there is no evidence of these species in Cayman. Their absence is noticeable because of their wide distribution and relative high abundance in most other countries with similar habitat and environmental conditions (for example in Jamaica/Belize/Turks and Caicos Islands/Bahamas [24], Gulf of Mexico/Florida [114], Florida/Bahamas [115], Bahamas [109,116,117], Columbia [113], Dutch Caribbean [35], Tobago [45], South Florida [118], US Virgin Islands [119], Turks and Caicos Islands [34], Belize [31]). However, these three species tend to be characteristic of continental margins, especially the bull shark which pups in estuarine waters [120–122], whereas in contrast Cayman is an isolated group of small islands in the center of the Caribbean Sea, and lacks any estuarine environments.

The relatively high abundance of Caribbean reef shark, nurse shark and hammerhead spp. in comparison to other species confirms studies of species abundances previously reported in Cayman [12,84,106] and in other Caribbean regions [23,35,45]. The distribution of species may depend on various factors such as sea temperature, salinity, habitat, prey availability and anthropogenic disturbances [23,24,38,46,123–129]. In contrast, some other species, especially apex predators (for example tiger, scalloped (*Sphyrna lewini*), and great hammerhead sharks (*Sphyrna mokarran*)), tend to occur at naturally lower abundances with corresponding low levels of residency as also suggested by others elsewhere [111,123,130–132]. In addition, however, the very largest species of sharks and fish appear to be more cautious in approaching close enough to divers to be seen, whereas Caribbean reef and nurse sharks, although cautious, are more inquisitive.

As most divers were bound by recreational sport diver limits (e.g., maximum depth of 30 m) and most dives undertaken from a boat on the shallower part or outer edge of the fringing reef, species that prefer greater depths or a different habitat [131] were likely under-represented in this study. Notably, hammerhead spp. in this present study. were usually encountered along the main wall; however, individuals were typically encountered swimming at a distance to the reef, making encounters with divers less likely than for reef-associated species (for example: nurse shark, Caribbean reef shark, lemon shark [46,124]).

## Abundance

The present study suggests that the populations of the species observed were relatively stable over the two-year period, a finding which was not unexpected given the time frame of the monitoring. Due to the life-history characteristics of sharks, including slow growth, late maturity, and low fecundity [133], for natural changes in shark populations to be detected, long-term monitoring over more than 10 years is generally required [134,135]. It is difficult to compare the present data with past abundance levels; however historic sighting records from Cayman (1960's–1980's [29,85]) suggest that over time encounters with hammerhead spp. have decreased, with Caribbean reef sharks may have increased, while those with nurse sharks may have been stable. A trend also observed by population estimates generated through photo identification and on BRUVS videos over a 4 year period [84]. Compared to data from BRUVS surveys (<30 m depth, 2009–2018) [12,84,106], hammerhead spp. appear to be more frequently encountered in this present study. This finding supports more recent research

(2022–2024) using deep-water BRUVS at depths of 50-200m [85], suggesting that numbers of hammerhead sharks in Cayman may recently have begun to recover. The status of shark populations in the Caribbean in the early 2000s was generally poorly known [136]. There were some areas where sharks were strongly depleted in the last half century (for example: Dutch Caribbean [98]) whereas coastal shark populations seem to have been stable in some other Caribbean regions over the past 10 years (Belize: 2000–2004 [31], 2000–2013 [137] and Bahamas: 2004–2015 [116]).

## Sex ratios

According to diver observations, the majority of sexed Caribbean reef and nurse sharks in Cayman were female and only female hammerhead spp. were reported. This is surprising considering other data indicate uniform sex ratios in Caribbean reef and nurse sharks in Cayman [30,84] and Belize [31]. A similar demographic feature was reported by SCUBA divers in grey reef sharks (*Carcharhinus amblyrhynchos*) from Sudan, and assumed to reflect observer bias [77]. The difficulty of sexing individuals observed in deeper water (i.e., viewed from above) or that are immature and so lack clearly visible claspers (especially nurse sharks) will likely explain, at least in part, the relative low number of reported male sharks.

Even so, the data could also be explained by partial sexual segregation by habitat and/or depth (with females tending to make more use of shallow coastal reefs), a phenomenon which is recognized in many species, particularly of coastal sharks [75,138–142].

While recognizing that there will have been some observer error in size estimates, it appears that the majority of Caribbean reef and hammerhead spp. were classified as mature but the majority of nurse sharks as immature. This suggests some degree of habitat partition based on maturity stage also recorded by BRUVS [84]. It seems likely that while immature nurse sharks seek shelter in shallow lagoonal waters, immature Caribbean reef sharks take refuge in depth. In some species segregation of life stages is thought to reflect differences in site-fidelity, habitat preference, energy requirement, or foraging strategy [36,143,144], as well as predator avoidance by smaller sharks [83,145–147].

## Spatial distribution of sharks

The results obtained through the Sharklogger Network are in accord with the habitat and depth use of Caribbean reef and nurse sharks as observed in Cayman using BRUVS, scientific long-lining and acoustic telemetry [12,30,83,84] and as recorded for these species elsewhere in the Caribbean [31,33,90,96,145]. For Caribbean reef sharks, the predominance of records from deeper reef areas (> 20 m depth), typical around the outer edge of the drop off, is consistent with observations of that this species spends time at depths over 100 m [31,33,137] thought to be related to foraging [36]. Nurse sharks in contrast occurred through a range of habitats and depths, with a greater relative abundance both in very shallow water and lagoons (<5 m depth) and at deeper depth (36–40 m) on the reef edge, reflecting the known habitat preferences of this species [138,148,149].

The results indicate that the three shark species for which data were analyzed were more abundant on particular islands and in specific areas. This is likely due to the occurrence of more favorable conditions related to sea temperature, habitat, prey availability and human disturbance in some areas as opposed to others, such as has been observed for several coastal shark species [32,150–152]. In this study, areas with relative high shark abundances, such as the N, NE, and SE of Grand Cayman, as well as the south side of Cayman Brac, are typically subject to more frequent and stronger current flow, prevailing wind and currents being from the north-east and east. Higher abundances in areas exposed to stronger current has also

been observed in a wide range of species [153–155], including Caribbean reef sharks [156], it is assumed because these areas tend to offer higher prey abundances as a result of a greater influx of nutrients and abundance of prey [19,157].

The relatively low abundances of sharks on the west coast could also be the result of individuals avoiding human disturbances such as fishing activities, habitat loss and pollution, as has been observed in some other countries [32,128,158]. Even though Cayman's largest MPA covers 9 km² of the coastal shelf on the west coast, providing effective protection from fishing [13,159], this coast is exposed to the highest levels of human activity, because it is where the majority of hotels and other tourist facilities are concentrated. Apart from people bathing along the beaches, there is considerable boat traffic, including of cargo vessels, cruise ships, and recreational fishing and diving vessels, such as is known to effect coastal sharks negatively [32,152]. This present study supports previous research using BRUVS, acoustic telemetry, catch-and-release and photo-ID that also found shark relative abundances to be higher outside MPAs than inside [12,30,83]

The data provide evidence of spatial segregation of Caribbean reef and nurse sharks since areas with high abundance of one species tended to have low abundances of the other species. This supports data obtained during related research using BRUVS [30] and acoustic telemetry [83]. Spatial segregation of sympatric, such as observed in this study, is common in reef fish [160] including sharks, as well as in most animal groups, and reflects habitat partitioning as species compete to secure limited resources of food and space [119,146,161,162].

While in general sharks may avoid areas of excessive human disturbance it is worth noting that for both Caribbean reef and nurse sharks, some individuals were recorded to appear on the first and second dive at two separate dive sites dived on the same day within about 45-60 min between dives. This usually occurred when the boat moved to an adjacent dive site, a few hundred meters away from the first dive site. It is evident that the effect of human disturbance may vary with the nature and degree of the activity and individual shark involved.

## Home range and site-fidelity

Analysis of home range size and site-fidelity were only possible for Caribbean reef and nurse sharks and confirmed the relatively small home range sizes and high site-fidelity of at least some individuals of both these species previously reported in Cayman [83,84]. In general Caribbean reef sharks appeared to be less site-attached, so it was surprising that data from divers indicated that on average individuals of this species had smaller home range sizes than did nurse sharks. Nevertheless, some individuals of both species were recorded by divers as having moved between dive sites up to 10 km apart. Comparable home range sizes have been reported in other studies [33,38,84,96,163]. It has been anticipated that home range size might be larger for Caribbean reef sharks than nurse sharks [33], but a more recent study found that nurse sharks to be the least site-attached species among various reef sharks [38,84]. Nurse sharks tend to rest by day among or near reef structures, but they also actively patrol individual home ranges [138].

Despite their capacity for long-distance travel across open ocean of both species as also observed by others [33,35,164], in this present dataset no individual shark was reported from more than one island. However, our acoustic telemetry data from Cayman has shown that some individual Caribbean reef sharks do move between islands [12,83]. The lack of records of inter-island movements in this study is likely due to these individuals spending only limited time on other islands or doing so away from an established dive site.

## Effect of marine protected areas

Despite extensive evidence that MPAs can benefit coastal shark populations [137,165,166], the results of this study indicate that in the Cayman Islands shark abundances are not higher

inside MPAs. Since MPAs are known to especially benefit species with relatively small home ranges and high site-fidelity [137,156,167], it was not surprising that hammerhead spp. with typically larger home ranges and even long-distance migrations [168], were found in the present study to be more abundant outside of MPAs. It was less expected that although Caribbean reef and nurse sharks appear to have relatively small home ranges and high site-fidelity, only nurse sharks, and not Caribbean reef sharks, were found to be more abundant inside MPAs. This confirms the findings from BRUVS surveys [12,30] and acoustic telemetry [83] that indicated that in the Cayman Islands sharks tend to be more abundant in areas that do not have protected area status. This is likely in part due to the fact that Cayman's MPAs were originally designed to protect areas favored by recreational water activities, combined with a tendency of Caribbean reef sharks to avoid human disturbance. It is also possible that other factors, such as environmental conditions, may override the effect of protection within MPAs or MPAs will have lesser to no effect on protected species if they are not threatened outside the areas. Nevertheless, as suggested by previous studies [12,83,84], this present study provides the evidence that at least a proportion of Caribbean reef sharks make occasional excursions well outside their core home range and the boundaries of any local MPA, making them more vulnerable to recreational fishing activities and unintended capture.

## Temporal patterns in abundance of sharks

For Caribbean reef sharks, the apparent decrease in relative abundance over summer (April–September), particularly in May and June, may reflect a seasonal habitat shift and/or greater mobility of individuals. Seasonal shifts in habitat use and increased mobility have been linked to prey availability or reproductive activity in various shark species [39,40,132]. In this study there was little evidence for seasonal change in foraging behaviour, but the observed seasonal variation in observed courtship and mating behaviour indicates that this change in abundance might be related to reproduction. Similar changes in behaviour were recorded by acoustic telemetry in Caribbean reef sharks [83].

In this study, the observations of courtship behaviour and mating in mature individuals, as well as the presence of very small (<1 m) Caribbean reef sharks, indicate that both mating and pupping in Caribbean reef sharks occurs between June and September, with a peak in August, as suggested by Ormond et al. [12] and also reported from other Caribbean regions [169,170]. While such firsthand observations are not common, more reports and images of reproductive behaviour have become available online (e.g., on YouTube: https://www.youtube.com/watch?v=ba3wlEprYzw, https://www.youtube.com/watch?v=E29I83Wkr5A). The mating and courting observations reported in this study were verified by the coordinator through personal communication with the relevant diver, and on one occasion, through firsthand personal observation by JK. The north coasts of both Grand and Little Cayman seem to be areas where both mating and pupping may occur. While no reproductive behaviour was reported from Cayman Brac, this might be due to a lower observer effort (Cayman Brac had the lowest number of participants and number of dives than the other two islands). Besides observations of mating behaviour, mating related scarring on females was reported when individuals returned to their usual home range at the end of summer. That pupping and mating occurs during the same season has also been observed in studies of other species [154,171–173].

For nurse sharks, the changes in abundance, with a peak in shark encounters in August, may also be related to reproduction. Nurse sharks were not observed mating, but very small (<1 m) nurse sharks were reported all year around from all islands and a pregnant female was observed June on Cayman Brac. Besides confirming that nurse sharks reproduce in Cayman, these observations suggest that pupping may occur throughout the year, but probably peaks in

summer, as reported for other regions [90]. Seasonal migrations linked to reproduction have been observed in nurse sharks with males returning every year to the same area during the mating season [138].

Besides reports of injuries suggestive of mating, divers also recorded that some individual Caribbean reef and nurse sharks had acquired minor to serious injuries while still within their home range. Notably, one nurse shark was found with serious cuts on the head and body. As the shark remained in the same vicinity for a few weeks, divers were able to document the healing process over time and even the health of the animal post recovery, since the individual was reported from adjacent dive sites after the wounds had healed. This case shows that reports from divers can be used to monitor the health of individuals and document the effects of injuries, however caused.

## Programme management

This study demonstrates that recreational divers can be used to monitor coastal shark populations in a highly cost-effective manner, providing a statistically robust alternative to traditional surveys. However, the development and supervision of such a citizen science program requires considerable time and effort. The success of this long-term program depended on effective recruitment, training, and close guidance of participants, as well as on careful management of datasets to ensure data quality. The recruitment process also required constant effort as for a variety of reasons divers frequently left the program. Sometimes, after initial interest, potential participants realized that their paid employment did not leave them enough time for the regular effort required. Other reasons for ending participation included pregnancy, injuries, equipment difficulties, leaving the island, change of jobs, and loss of interest. The number of dropouts was highest in September and October, when simultaneously the rate of recruitment was lowest, probably because the turn-over of staff in the local workforce is highest at that time of year.

It was of note that while most divers quickly understood the process of logging shark sightings, some participants had difficulty in understanding that all dives needed to be reported. Many conservation minded divers might have previously participated in or heard of the general concept of citizen science programs organized by widely known organizations such as REEFBASE (https://worldfishcenter.org/publication/reefbase-global-database-coral-reef-systems-and-their-resources), REEF (https://www.reef.org/programs/volunteer-fish-survey-project), NOAA (https://oceanservice.noaa.gov/citizen-science/) or PADI AWARE (https://blog.padi.com/what-is-citizen-science/) which require the record of only shark sightings. In this study, often it took careful explaining of the relevant science before participants would submit records of dives without shark sightings. It also took time for the coordinator to set up a logging protocol that streamlined the process for dive operators. As dive staff have typically long and busy working days, unless the dive owner or manager were supportive of their participation, many dive staff lost interest quite quickly.

On-island leadership, encouragement and close contact were required to ensure that participants remained engaged in the program and maintained regular sampling throughout the study period, as described for other citizen science initiatives [72,174,175]. This was achieved by offering various incentives such as public talks, public recognition of divers, appreciation events for divers, personal meetings, one-on-one dives, annual data reviews of personal diving data and preliminary results (for business websites and personal use), participant dive get-togethers, monthly newsletters (containing fieldwork insights and results of ongoing shark research conducted by DoE/MCI) and opportunities to join field trips. In this way it proved possible to keep the majority of participants interested in the program, as also reported by others [174].

In conclusion, this present study has generated novel invaluable information on the abundance, distribution, movement patterns and reproductive behaviour of the commonest reef shark species in the Cayman Islands with robust results that are comparable with those from BRUVS [12,84] and acoustic telemetry [83]. It has also demonstrated that a long-term citizen science programme involving a large number of participants and effort-based data collection can be an affordable, non-invasive, and repeatable tool with potential applications in other regions, especially for projects with limited financial resources. Aside from scientific objectives, this study also demonstrated the potential for engaging local communities and stakeholders to benefit local shark research. The direct involvement of the recreational diving community in shark research has the potential to increase both the socio-economic value of sharks and benefit shark conservation objectives.

## Supporting information

**S1 Fig. "Introduction Package" provided to participants in the Sharklogger Network.** The package contains relevant information for the data collection.
(PDF)

**S2 Fig. Examples of submitted Shark Logs by participants using A) handwritten PDF, B) typed PDF and C) Microsoft Excel format.** Personal information has been redacted.
(TIF)

**S3 Fig. Map showing the distribution of diving effort by participants of the Sharklogger Network in the Cayman Islands during 2017 + 2018.** Circles indicate the location of dive sites accessible by boat or from shore, the circle size is scaled by the diving effort (number of dives). Dive sites with a greater number of dives are indicated by larger circles than dive sites with fewer dives. The color of circles and corresponding range of diving effort is shown on map. The line around each island indicates the 25m (80ft) depth contour and the shaded areas indicate the extent of MPAs. Created by the Department of Environment, Cayman Islands Government. Insert layer's geography was developed by Esri and sourced from Garmin International, Inc., the U.S. Central Intelligence Agency (The World Factbook), and the National Geographic Society for use as a world basemap [82].
(TIF)

**S1 Table. Sample protocol for divers to record data during dives as part of the Sharklogger Network in the Cayman Islands during the study period (2017 + 2018).**
(PDF)

**S2 Table. Reference list of identified Caribbean reef sharks (C-number, n = 11) and nurse sharks (N-number, n = 16) recorded by divers from the Sharklogger Network in the Cayman Islands during the study period (2017 + 2018).**
(PDF)

**S3 Table. Summary of number of shark logs that included information on variables: maximum depth, current strength, temperature, dive duration, lionfish culling, visibility, and dive group size.**
(PDF)

**S4 Table. Post-hoc Dunn test results of pairwise comparison of relative abundance between shark species.** Test statistic (Z) and p-values are reported and significant differences, at the 0.05 level, are marked with *.
(PDF)

**S5 Table. Post-hoc Dunn test results of pairwise comparison of relative abundance of sharks (n = 8 species) and of nurse sharks between the Cayman Islands.** Test statistic (Z) and p-values are reported and significant differences, at the 0.05 level, are marked with *.
(PDF)

**S6 Table. Post-hoc Dunn test results of pairwise comparison of relative abundance between areas on Grand Cayman.** Test statistic (Z) and p-values are reported and significant differences, at the 0.05 level, are marked with *
(PDF)

**S7 Table. Post-hoc Dunn test results of pairwise comparison of relative abundance between areas on Little Cayman and Cayman Brac.** Test statistic (Z) and p-values are reported and significant differences, at the 0.05 level, are marked with *.
(PDF)

**S8 Table. Post-hoc Dunn test results of pairwise comparison of relative abundance between months.** Test statistic (Z) and p-values are reported and significant differences, at the 0.05 level, are marked with *.
(PDF)

**S9 Table. Post-hoc Dunn test results of pairwise comparison of relative abundance between depth.** Test statistic (Z) and p-values are reported and significant differences, at the 0.05 level, are marked with *.
(PDF)

**S10 Table. Post-hoc Dunn test results of pairwise comparison of relative shark abundance between current strength groups.** Test statistic (Z) and p-values are reported and significant differences, at the 0.05 level, are marked with *.
(PDF)

**S11 Table. Minimum linear displacement (MLD) and site-fidelity (SFI) of Caribbean reef sharks (n = 11) and nurse sharks (n = 16) identified by reports from the Sharklogger Network in the Cayman Islands during the study period (2017 + 2018).**
(PDF)

**S12 Table. Summary of shark behaviour derived from comments of dives (n = 24,442) reported by the Sharklogger Network in the Cayman Islands during the study period (2017 + 2018).**
(PDF)

## Acknowledgements

We thank Gina Ebanks-Petrie, DoE Director, for her support of this initiative. We especially thank all the citizen science participants (see full list www.doe.ky) who volunteered their time and collected data during numerous dives and without whom this study would not have been possible. We also thank Kat Mason, Anne Veeder, Megan Ehman, Enno Krebbers and Joy Mulholland for their assistance with the data collection and processing.

## Author contributions

**Conceptualization:** Johanna Kohler, Mauvis Gore, Rupert Ormond, Timothy Austin.

**Data curation:** Johanna Kohler.

**Formal analysis:** Johanna Kohler.

**Funding acquisition:** Mauvis Gore, Rupert Ormond, Timothy Austin.

**Investigation:** Johanna Kohler.

**Methodology:** Johanna Kohler.

**Project administration:** Johanna Kohler.

**Resources:** Johanna Kohler, Timothy Austin, Jeremy Olynik.

**Software:** Johanna Kohler, Jeremy Olynik.

**Supervision:** Mauvis Gore, Rupert Ormond, Timothy Austin.

**Validation:** Johanna Kohler.

**Visualization:** Johanna Kohler, Jeremy Olynik.

**Writing – original draft:** Johanna Kohler.

**Writing – review & editing:** Johanna Kohler, Mauvis Gore, Rupert Ormond, Timothy Austin, Jeremy Olynik.

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
