## [Decision Letter · Decision Letter 0]

29 Dec 2024

PONE-D-24-17420The Sharklogger Network – monitoring Cayman Islands shark populations through an innovative citizen science programPLOS ONE

Dear Dr. Kohler,

Thank you for submitting your manuscript to PLOS ONE. After careful consideration, we feel that it has merit but does not fully meet PLOS ONE’s publication criteria as it currently stands. Therefore, we invite you to submit a revised version of the manuscript that addresses the points raised during the review process.

Your manuscript was favorably reviewed by two experts in the field. However, I believe it would be beneficial and improve your paper if you addressed the minor yet important comments of reviewer 2.

We look forward to receiving your revised manuscript.

Kind regards,

Ulrike Gertrud Munderloh, Ph.D.

Academic Editor

PLOS ONE

Journal Requirements:

4. We note that you have referenced (ie. Bewick et al. [5]) which has currently not yet been accepted for publication. Please remove this from your References and amend this to state in the body of your manuscript: (ie “Bewick et al. [Unpublished]”) as detailed online in our guide for authors

Reviewers' comments:

Reviewer's Responses to Questions

**Comments to the Author**

1. Is the manuscript technically sound, and do the data support the conclusions?

Reviewer #1: Yes

Reviewer #2: Yes

2. Has the statistical analysis been performed appropriately and rigorously? 

Reviewer #1: I Don't Know

Reviewer #2: Yes

3. Have the authors made all data underlying the findings in their manuscript fully available?

Reviewer #1: Yes

Reviewer #2: Yes

4. Is the manuscript presented in an intelligible fashion and written in standard English?

Reviewer #1: Yes

Reviewer #2: Yes

5. Review Comments to the Author

Reviewer #1: On the whole, the paper was well-written, especially the Abstract and Discussion sections, clearly laying out the goals and objectives of the study. Also, I liked that the protocol used so-called 'zero data,' an important tool in fixed survey areas with known species, something many shark surveys lack.

Possible concerns:

1. The culling of lionfish was carried out in some cases with the simultaneous counting of shark species in the area and the author admits that this might have affected the natural behavior of the sharks and possibly attracting species which would not normally be in that area.

2. Author also reports some sharks following divers from one count site to another, increasing the likelihood of double-counting, also affecting natural migration/behavior patterns.

3. Some surveys were conducted in MPAs, which the author claimed had no bearing on the relative abundance of certain species, despite the ban restrictions on fishing in those areas--however, separate MPA species surveys would need to be done to determine if this was a factor.

4. Mating behavior was observed, which is extremely rare for casual or even scientific divers to witness.

6. Study did not use underwater cameras, so species identification was dependent upon proper diver training in visual species identification.

Reviewer #2: This manuscript details an exciting contribution of citizen science to shark conservation research. The study is conducted in a unique location wherein this type of citizen science can reliably be harnessed year-round and with high frequency. The results demonstrate how citizen science, when implemented in a controlled manner, can substantially add to our current understanding of shark ecology and can contribute to conservation efforts.

My only major comments are: 1. The authors should provide more information on how many dives occurred in each location. Can you use circle size to demonstrate this in Fig. 1? This can align with more discussion on how effort is related to sightings. 2. How does this relate to other citizen science programs - not necessarily related to sharks? The authors mention previous shark-focused efforts, but an evaluation of their program's success/concerns may be more rigorous in comparison with other citizen science programs for other species.

6. PLOS authors have the option to publish the peer review history of their article (what does this mean? ). If published, this will include your full peer review and any attached files.

**Do you want your identity to be public for this peer review?** For information about this choice, including consent withdrawal, please see our Privacy Policy .

Reviewer #1: **Yes: ** Michael Bear

Reviewer #2: No

---

## [Author Response · Author response to Decision Letter 0]

31 Jan 2025

Thank you for the opportunity to revise our manuscript and for forwarding the comments from the reviewers. In response to the feedback the following amendments have now been made to the manuscript. Please note that the line numbers below refer to the line numbers in the unmarked version of the revised manuscript.

1. The manuscript was reviewed, and amendments were made to meet the journal’s style requirements referencing the templates provided (e.g. double-spacing, headings, file names)

2. Additional information were provided in the Methods section to clarify that no permits were required. (line 368-377, also see point #3 below)

3. The full ethics statement was included in the Methods section and additional information is now provided to clarify that no permits were required. (line 368-377)

4. The reference Bewick et al [5] was not found in the manuscript. Please advise what you meant by “Bewick et al. [5]”. In the original manuscript, we did find Reference item #160 Williamson et al. 2020 which met the “unpublished” criteria. This reference was therefore removed in text and from the Reference list.

5. Reference list was amended to reflect the removal of Williamson et al. 2020 because this reference does not meet the journals requirements (see point 4 above). It was not replaced because we did cite others which are sufficient. Refence list was checked against citation in manuscript.

We appreciate the positive feedback from Reviewer #1 about the rigorous protocol and overall manuscript. Thank you for raising concerns and we made changes to clarify each point raised as detailed below:

1. The recording of diver behaviour with regards to lionfish culling was meant to give additional information in case a specific trend was observed (e.g. more shark on specific dives/outliers) and to quantify the dive behaviour of local divers (how much lionfish culling occurred and in what areas). In fact, it was only on a relatively small number of dives that lionfish culling occurred (1,286 dives, 5.6 % of total dives), and these accounted for only a small proportion of shark sightings (338 shark sightings, 7.8 % of total shark sightings). As the reviewer mentioned, this is highlighted in the Results and Discussion to provide the reader with the relevant context to interpret the results.

While it is entirely possible that the lionfish culling might have attracted species which would not normally be in that area, as the reviewer pointed out, our observations show that sharks seemed to be more visible on lionfish culling dives (i.e. come closer to divers and hence into the diver’s field of view) but the species recorded were no different than observed on other occasions. We believe that the difference in ‘visibility of sharks’ might slightly bias the spatial distribution of relative abundance of a species but not influence species composition in an area. We acknowledge the concern and have expanded this section in the Discussion to make it clearer (line 678-683).

2. Thanks for raising this point. Natural migrations/behaviour patterns were not affected as we are referring to movement between adjacent sites <0.3 km apart. Also the sharks do not actually “follow divers”, perhaps a better way of explaining it would be that sharks “reappear at a second, adjacent site (usually about a few hundred meters apart) following movement of the dive boat”. This would happen normally since in most such cases adjacent dive sites would be within an individual shark’s home range. Hence, such observations do not exaggerate the counts and would not classify as “double-counting” because while it is the same shark being seen, it is at two separate sites/dives. The risk of double counting of the same shark at the same site/dive was minimized by taking the number reported by the dive leader after consultation with the dive group (not by adding the counts of separate divers that may all have reported the same shark(s). In addition, various steps were taken during data validation/verification by the coordinator during data cleaning and management (detailed in Data analysis and Discussion). Nevertheless, we appreciate the concern raised by the reviewer and have amended the relevant sections in the Results and Discussion to make the situation clearer (line 590-591, and 851-856). It is also of note that this did not happen on a regular basis but only with a few individual sharks “resident” individuals often known to divers because of their regular occurrence. In short, we don’t believe that such shark behaviour will have impacted the results.

3. Thanks for highlighting the fact that additional references may be useful. Despite other papers on the MPA effect, for sharks in Cayman this seems not to be true. All of our surveys since 2009 show that MPAs don’t harbor greater numbers of sharks than areas outside MPA boundaries. This may be partly because in Cayman original MPAs were designed to “protect areas with high diving pressure” not areas with high biodiversity (briefly mentioned in Introduction line 72-75), hence shark abundance and movements were not taken into consideration when MPAs were established in 1986. In addition, our research has demonstrated that the local sharks regularly move over home ranges greater than the areas covered by MPAs. This is so even though most MPAs in Cayman are believed to have higher prey availability (based on knowledge on currents, waves, other fish abundances and distribution of fishing activities) (mentioned in Discussion starting line 825-834, 835-842, 883-890). We added references to previous research where appropriate as well as a few sentences to the Discussion to make the situation clearer (Discussion line 843-844, 891-893).

4. Thanks for appreciating the significance of these observations. Indeed, these firsthand observations are not common, although as an internet search for papers and images will show (links now added to the Discussion for reference), have become much commoner in recent years. One of these I observed myself (courting behaviour) and I was literally holding my breath because I thought I was about to see a mating event in real life! That case apart, all of the reports of mating were verified through personal communication to double-check what exactly the divers observed. Since the completion of this study, there have been many more reports of mating sharks being observed by local divers, always around the same time period and areas. Such observations used to be considered unusual but have become more common globally as well as in Cayman. This is likely because more divers (and more divers that are trained/aware) are diving in more places where sharks occur. We have added a reminder that this data was validated by the lead author to the relevant section in the Results and Discussion to make it clearer (line 596-599, 906-912).

5. We did encourage the use of cameras by participants, and some divers did use cameras to verify shark species, injuries and behaviour. However, most recorders were dive center leaders, and dive centers have safety protocols that do not allow for camera use by dive staff during dives. To minimize potential misidentifications, any identifications (and numbers) which were questionable, or about which divers had concerns, were checked by the coordinator who would validate through personal communication including referencing pictures, if necessary (detailed in Sample Protocol line 83-92, 127-142). Overall, the data fully support previous scientific surveys using cameras and catch-and-release in Cayman (Ormond et al. 2017, Kohler et al. 2022, 2023), therefore it is reasonable to assume that the training was sufficient to enable divers to identify most shark species correctly. Thanks for raising this point; we have amended the section in the Results and Discussion to make the situation clearer (596-598, 699-708, 728-730, 737-744, 787-789).

We likewise thank Reviewer #2 for the positive feedback regarding the benefits of this manuscript for citizen science and shark conservation research. Both suggestions made by Reviewer #2 have been taken into consideration and amendments to the manuscript have been made as follows:

1. We appreciate the suggestion of adding more information about the spatial diving effort of participants across all three islands. We have this information but initially did not include it in the manuscript in order to keep the paper concise. We agree with Reviewer #2 that the diving effort will aid the reader to better understand the results. We have therefore added a detailed map to the Supporting Information and cited it at the beginning of the Results (line 383-384). We have also amended the relevant section in the Discussion to make this point clearer (line 666-667).

2. We thank the reviewer for pointing out the benefit of expanding on this. More details about the study’s limitations have been added where relevant in the Discussion (e.g. line 669-683). Other sections of the Discussion have been amended to include comparisons with other citizen science programs (E.g. 699-708, 737-744, 946-953).

---

## [Editor Report · Decision Letter 1]

6 Feb 2025

The Sharklogger Network – monitoring Cayman Islands shark populations through an innovative citizen science program

PONE-D-24-17420R1

Dear Dr. Kohler,

We’re pleased to inform you that your manuscript has been judged scientifically suitable for publication and will be formally accepted for publication once it meets all outstanding technical requirements.

Kind regards,

Ulrike Gertrud Munderloh, Ph.D.

Academic Editor

PLOS ONE
---

## [Editor Report · Acceptance letter]

PONE-D-24-17420R1

PLOS ONE

Dear Dr. Kohler,

I'm pleased to inform you that your manuscript has been deemed suitable for publication in PLOS ONE. Congratulations! Your manuscript is now being handed over to our production team.

Kind regards,

on behalf of

Dr. Ulrike Gertrud Munderloh

Academic Editor

PLOS ONE